# Generalization Analysis for Label-Specific Representation Learning

**Yi-Fan Zhang**[1,3]**, Min-Ling Zhang**[2,3*]

[1] School of Cyber Science and Engineering, Southeast University, Nanjing 210096, China
[2] School of Computer Science and Engineering, Southeast University, Nanjing 210096, China
[3] Key Laboratory of Computer Network and Information Integration (Southeast University),
Ministry of Education, China
{yfzh, zhangml}@seu.edu.cn

## Abstract

Label-specific representation learning (LSRL), i.e., constructing the representation with specific discriminative properties for each class label, is an effective strategy to improve the performance of multi-label learning. However, the generalization analysis of LSRL is still in its infancy. The existing theory bounds for multi-label learning, which preserve the coupling among different components, are invalid for LSRL. In an attempt to overcome this challenge and make up for the gap in the generalization theory of LSRL, we develop a novel vector-contraction inequality and derive the generalization bound for general function class of LSRL with a weaker dependency on the number of labels than the state of the art. In addition, we derive generalization bounds for typical LSRL methods, and these theoretical results reveal the impact of different label-specific representations on generalization analysis. The mild bounds without strong assumptions explain the good generalization ability of LSRL.

## 1 Introduction

Multi-label learning has received continued attention in machine learning community due to its widespread encounters in real-world applications, where each object is represented by a single instance associated with multiple class labels [Zhang and Zhou, 2014, Liu et al., 2022]. The goal of multi-label learning is to model real-world objects with multiple semantics. It has made important advances in multimedia content annotation [Cabral et al., 2011, You et al., 2020], text categorization [Rubin et al., 2012, Xun et al., 2020], bioinformatics [Cesa-Bianchi et al., 2012, Chen et al., 2017] and other fields [Yu et al., 2005]. The failure to take into account that each class label may possess its own discriminative properties results in the most straightforward strategy which exploits the identical representation of an instance for dealing with multi-label data being perhaps suboptimal [Zhang and Wu, 2015, Huang et al., 2016, Hang and Zhang, 2022]. In recent years, label-specific representation learning (LSRL) [Zhang and Wu, 2015] has been proposed to facilitate the discrimination of each class label by tailoring its own representations. Due to its ability to model distinct characteristics for each class label, LSRL has become an effective strategy to improve the performance of multi-label learning [Huang et al., 2015, 2016, 2018, Yu and Zhang, 2022, Hang and Zhang, 2022, Hang et al., 2022]. Although LSRL has achieved impressive empirical advances in multi-label learning, the problem of understanding LSRL theoretically remains completely under-explored.

In recent years, efforts to explain why multi-label models generalize well is an important open problem in multi-label learning community. The empirical success of LSRL makes the generalization

---

*Corresponding author

analysis of LSRL an important problem in multi-label learning. However, existing theoretical results and analysis methods are not applicable to LSRL, which leads to a serious lack of progress in its generalization analysis. A satisfactory and complete study of the generalization analysis for LSRL should include two aspects: 1) the dependency of the generalization bounds on the number of labels (i.e., $c$) should be weaker than square-root, and 2) the relationship among components should be decoupled in the generalizability analysis. First, existing bounds with a linear or square-root dependency on $c$ are difficult to explain empirical success of multi-label learning. For example, the bounds with a linear dependency on $c$ are vacuous (i.e., no longer less than 1) for commonly used multi-label datasets such as CAL500, corel5k, rcv1-s1, Corel16k-s1, delicious, iaprtc12, espgame, etc., since $c$ is larger than $\sqrt{n}$ (the number of examples) for these datasets. The bounds with a linear or square-root dependency on $c$ are vacuous for commonly used extreme multi-label datasets such as Wiki 10, Amazon-670K, etc., since $c$ is larger than $n$. The above failure of existing bounds for multi-label learning also applies to LSRL. Hence, this suggests that only the bounds with weaker dependency on $c$ can provide effective theoretical guarantees. Second, existing theoretical results preserve the coupling among different components [Lei et al., 2015, Wu and Zhu, 2020, Wu et al., 2021a,b]. However, LSRL decomposes the multi-label learning problem into multiple binary classification problems, which means that the relationship among different components needs to be decoupled in the generalization analysis. Hence, we need to develop new analysis methods for LSRL. As a matter of fact, theoretical research on LSRL can also promote a better understanding of multi-label learning.

In this paper, we derive novel and tighter bounds based on the Rademacher complexity for LSRL. Specifically, we develop a novel vector-contraction inequality with the assumption that the loss function is Lipschitz continuous w.r.t. the $\ell_\infty$ norm, then we derive the bound for general function classes of LSRL with no dependency on $c$, up to logarithmic terms, which is tighter than the state of the art. In addition, we also analyze the bounds for several typical LSRL methods, and we show that the construction method of label-specific representations will affect the generalization bound.

Our bounds for LSRL improve the dependency on $c$. Major contributions of the paper include:

- We develop a novel vector-contraction inequality for $\ell_\infty$ norm Lipschitz continuous loss, which overcomes the limitations of existing theoretical results and provides a theoretical tool for the generalization analysis of LSRL.

- We derive bounds for general function classes of LSRL with a weaker dependency on $c$ than the state of the art, which provides a general theoretical guarantee for LSRL.

- We derive bounds for typical LSRL methods, which reveals the impact of different label-specific representations on the generalization analysis. The theoretical techniques and results on $k$-means clustering, Lasso, and DNNs involved here may be of independent interest.

## 2 Related Work

### 2.1 Multi-Label Learning

Multi-label learning is one of the most studied and important machine learning paradigms in practice [Zhang and Zhou, 2014, Liu et al., 2022]. To cope with the challenge that the output space is exponential in size to the number of class labels, modeling label correlations is adopted as a feasible strategy to facilitate the learning process. Generally speaking, existing methods can be roughly grouped into three major categories based on the order of label correlations being considered, namely first-order methods [Boutell et al., 2004, Zhang and Zhou, 2007, Zhang et al., 2018] which tackle multi-label learning problem by decomposing it into a number of independent binary classification problems, second-order methods [Elisseeff and Weston, 2001, Zhu et al., 2018, Sun and Zhang, 2021] which tackle multi-label learning problem by considering pairwise relationships between labels, and high-order methods [Ji et al., 2010, Xu and Guo, 2021] which tackle multi-label learning problem by exploiting high-order relationships among labels. Recent years, benefiting from the good generalization performance of deep learning, deep methods such as recurrent neural networks [Yazici et al., 2020], graph neural networks[Chen et al., 2020], and embedding models [Bai et al., 2020, Dahiya et al., 2021] have been explored to model label correlations.

As a complement to the exploitation of label correlations, label-specific representation learning (LSRL) have been proven to be another effective strategy to improve multi-label learning by manipulating the input space. Existing methods can be roughly grouped into three major categories

based on the construction method of label-specific representations, i.e., prototype-based label-specific representation transformation methods [Zhang and Wu, 2015, Zhang et al., 2015, Weng et al., 2018, Guo et al., 2019, Zhang and Li, 2021] which generate label-specific representations by treating the prototypes of each class label as the transformation bases, label-specific feature selection methods [Huang et al., 2015, 2016, 2018, Weng et al., 2020, Yu and Zhang, 2022] which generate label-specific representations by retaining a feature subset as the most pertinent features for each class label, and deep label-specific representation methods [Hang and Zhang, 2022, Hang et al., 2022] which learn label-specific representations in an end-to-end manner by exploiting deep models. In particular, $k$-means clustering-based LSRL, i.e., LIFT [Zhang and Wu, 2015], is the representative method of prototype-based label-specific representation transformation methods, which constructs label-specific representations by querying the distances between the original inputs and the cluster centers for each class label. Lasso-based LSRL, i.e., LLSF [Huang et al., 2016], is the representative method of label-specific feature selection methods, which presents a Lasso-based framework with the constraint of pairwise label correlations for each class label. DNN-based LSRL, i.e., CLIF [Hang and Zhang, 2022], is the representative method of deep label-specific representation methods, which proposes to learn label semantics and label-specific representations in a collaborative way. In this paper, we will analyze the generalization bounds of these three representative LSRL methods.

## 2.2 Generalization Bounds for Multi-Label Learning

Dembczynski et al. [2010] derived the relationship between the expectations of Hamming and Subset loss based on the regret analysis on Hamming and Subset loss, and Dembczynski et al. [2012] further performed regret analysis on Ranking loss, which provided preliminary theoretical insights for understanding Hamming, Subset and Ranking loss. With the typical vector-contraction inequality [Maurer, 2016] (i.e., assume that the loss function is Lipschitz continuous w.r.t. the $\ell_2$ norm), one can obtain generalization bounds of order $O(c/\sqrt{n})$ for multi-label learning. Wu and Zhu [2020], Wu et al. [2021a] showed that the order of the generalization bounds for Subset loss, Hamming Loss and reweighted convex surrogate univariate loss can be improved to $O(\sqrt{c/n})$, which preserved the coupling among different components and exploited the relationship between loss functions. Liu et al. [2018] also obtained a generalization bound of order $O(\sqrt{c/n})$ for the dual set multi-label learning, which was analyzed under the margin loss and kernel function classes.

Wu et al. [2021b] derived a generalization bound of order $O(\log(nc)/n\sigma)$ for norm regularized kernel function classes with the assumptions that the loss function is Lipschitz continuous w.r.t. the $\ell_\infty$ norm and the regularizer is $\sigma$-strongly convex with respect to some norms. Yu et al. [2014] obtained a generalization bound of order $O(1/\sqrt{n})$ for trace norm regularized linear function classes with Decomposable loss. Xu et al. [2016] used the local Rademacher complexity to derive a generalization bound of order $\widetilde{O}(1/n)$ for trace norm regularized linear function classes with the assumption that the singular values of the weight matrix decay exponentially. These theoretical results all preserved the coupling among different components. In addition, Wu et al. [2023] obtained $O(1/\sqrt{n})$ bounds for Macro-Averaged AUC and gave thorough discussions about its relationships with the label-wise class imbalance, which transformed the macro-averaged maximization problem in multi-label learning into the problem of learning multiple tasks with graph-dependent examples. Here we obtain $\widetilde{O}(1/\sqrt{n})$ bounds with the state-of-the-art dependency on the number of labels for LSRL function classes under the assumption that the loss function is Lipschitz continuous w.r.t. the $\ell_\infty$ norm.

## 3 Preliminaries

Let $[n] := \{1, \ldots, n\}$ for any natural number $n$. In the context of multi-label learning, given a dataset $D = \{(\boldsymbol{x}_1, \boldsymbol{y}_1), \ldots, (\boldsymbol{x}_n, \boldsymbol{y}_n)\}$ with $n$ examples which are identically and independently distributed (i.i.d.) from a probability distribution $P$ on $\mathcal{X} \times \mathcal{Y}$, where $\mathcal{X} \subset \mathbb{R}^d$ denotes the $d$-dimensional input space and $\mathcal{Y}$ denotes the label space with $c$ class labels, $\boldsymbol{x} \in \mathcal{X}$, $\boldsymbol{y} \in \mathcal{Y} \subseteq \{-1, +1\}^c$, i.e., each $\boldsymbol{y} = (y_1, \ldots, y_c)$ is a binary vector and $y_j = 1$ ($y_j = -1$) denotes that the $j$-th label is relevant (irrelevant), $j \in [c]$. The task of multi-label learning is to learn a multi-label classifier $\boldsymbol{h} \in \mathcal{H} : \mathcal{X} \mapsto \{-1, +1\}^c$ which assigns each instance with a set of relevant labels. A common strategy is to learn a vector-valued function $\boldsymbol{f} = (f_1, \ldots, f_c) : \mathcal{X} \mapsto \mathbb{R}^c$ and derive the classifier by a thresholding function which divides the label space into relevant and irrelevant label sets.

For any vector-valued function $\boldsymbol{f} : \mathcal{X} \mapsto \mathbb{R}^c$, the prediction quality on the example $(\boldsymbol{x}, \boldsymbol{y})$ is measured by a loss function $\ell : \mathbb{R}^c \times \{-1, +1\}^c \mapsto \mathbb{R}_+$. The goal of learning is to find a hypothesis $\boldsymbol{f} \in \mathcal{F}$ with good generalization performance from the dataset $D$ by optimizing the loss $\ell$. The generalization performance is measured by the expected risk: $R(\boldsymbol{f}) = \mathbb{E}_{(\boldsymbol{x}, \boldsymbol{y}) \sim P}[\ell(\boldsymbol{f}(\boldsymbol{x}), \boldsymbol{y})]$. We denote the empirical risk w.r.t. the training dataset $D$ as $\widehat{R}_D(\boldsymbol{f}) = \frac{1}{n} \sum_{i=1}^n \ell(\boldsymbol{f}(\boldsymbol{x}_i), \boldsymbol{y}_i)$. We denote the optimal risk as $R^* = \inf_{\boldsymbol{f} \in \mathcal{F}} R(\boldsymbol{f})$, the minimizer of the optimal risk as $\boldsymbol{f}^* = \arg\min_{\boldsymbol{f} \in \mathcal{F}} R(\boldsymbol{f})$ and denote the minimizer of the empirical risk as $\hat{\boldsymbol{f}}^* = \arg\min_{\boldsymbol{f} \in \mathcal{F}} \widehat{R}_D(\boldsymbol{f})$. In addition, we define the loss function space as $\mathcal{L} = \{\ell(\boldsymbol{f}(\boldsymbol{x}), \boldsymbol{y}) : \boldsymbol{f} \in \mathcal{F}\}$, where $\mathcal{F}$ is the vector-valued function class.

### 3.1 Label-Specific Representation Learning

Label-specific representation learning aims to construct the representation with specific discriminative properties for each class label to facilitate its discrimination process. The basic idea of LSRL is to decompose the multi-label learning problem into $c$ binary classification problems, i.e., decoupling the relationship among different components, where each binary classification problem corresponds to a possible label in the label space. We consider the prediction function for each label of the general form $f_j(\boldsymbol{x}) = \langle \boldsymbol{w}_j, \zeta_j(\phi_j(\boldsymbol{x})) \rangle$, where the inner nonlinear mapping $\phi_j$ corresponds to the nonlinear transformation induced by the construction method of label-specific representation, while the outer nonlinear mapping $\zeta_j$ refers to the nonlinear mapping corresponding to the classifier learned on the generated label-specific representation. We define a vector-valued function class of LSRL as follows:

$$
\begin{aligned}
\mathcal{F} = \{\boldsymbol{x} \mapsto \boldsymbol{f}(\boldsymbol{x}) : &\boldsymbol{f}(\boldsymbol{x}) = (f_1(\boldsymbol{x}), \ldots, f_c(\boldsymbol{x})), \\
&f_j(\boldsymbol{x}) = h_j\left(\phi_j(\boldsymbol{x})\right) = \boldsymbol{w}_j^\top \zeta_j(\phi_j(\boldsymbol{x})), \boldsymbol{w} = (\boldsymbol{w}_1, \ldots, \boldsymbol{w}_c) \in \mathbb{R}^{d \times c}, \\
&\alpha(\boldsymbol{w}) \leq \Lambda, \beta(\zeta_j(\cdot)) \leq A, \gamma(\phi_j(\cdot)) \leq C, \boldsymbol{x} \in \mathcal{X}, j \in [c], \Lambda, A, C > 0\}, \quad (1)
\end{aligned}
$$

where $\alpha$ represents a functional that constrains weights, $\beta$ represents a functional that constrains nonlinear mappings $\zeta_j$, $\gamma$ represents a functional that constrains nonlinear mappings $\phi_j$.

### 3.2 Related Evaluation Metrics

A number of evaluation metrics are proposed to measure the generalization performance of different multi-label learning methods. Here we focus on commonly used evaluation metrics, i.e., Hamming Loss, Subset Loss, Ranking Loss and Decomposable Loss. However, the above mentioned loss functions are typically $0/1$ losses, which are actually hard to handle in optimization. Hence, one usually consider their surrogate losses, which are defined as follows:

**Hamming Loss** : 
$$\ell_H(\boldsymbol{f}(\boldsymbol{x}), \boldsymbol{y}) = \frac{1}{c} \sum_{j=1}^c \ell_b(y_j f_j(\boldsymbol{x})),$$

where the base convex surrogate loss $\ell_b$ can be various popular forms, such as the hinge loss, the logistic loss, the exponential loss and the squared loss.

**Subset Loss** : 
$$\ell_S(\boldsymbol{f}(\boldsymbol{x}), \boldsymbol{y}) = \max_{j \in [c]} \{\ell_b(y_j f_j(\boldsymbol{x}))\}.$$

**Ranking Loss** : 
$$\ell_R(\boldsymbol{f}(\boldsymbol{x}), \boldsymbol{y}) = \frac{1}{|Y^+||Y^-|} \sum_{p \in Y^+} \sum_{q \in Y^-} \ell_b(f_p(\boldsymbol{x}) - f_q(\boldsymbol{x})),$$

where $Y^+$ ($Y^-$) denotes the relevant (irrelevant) label index set induced by $\boldsymbol{y}$, and $|\cdot|$ denotes the cardinality of a set.

**Decomposable Loss** : 
$$\ell_D(\boldsymbol{f}(\boldsymbol{x}), \boldsymbol{y}) = \sum_{j=1}^c \ell_b(y_j f_j(\boldsymbol{x})).$$

### 3.3 Related Complexity Measures

Here we introduce the complexity measures involved in theoretical analysis. The Rademacher complexity is used to perform generalization analysis for LSRL.

**Definition 1** (Rademacher complexity). *Let $\mathcal{G}$ be a class of real-valued functions mapping from $\mathcal{X}$ to $\mathbb{R}$. Let $D = \{\boldsymbol{x}_1, \ldots, \boldsymbol{x}_n\}$ be a set with $n$ i.i.d. samples. The empirical **Rademacher complexity** over $\mathcal{G}$ is defined by $\hat{\Re}_D(\mathcal{G}) = \mathbb{E}_{\boldsymbol{\epsilon}} \left[\sup_{g \in \mathcal{G}} \frac{1}{n} \sum_{i=1}^n \epsilon_i g(\boldsymbol{x}_i)\right]$, where $\epsilon_1, \ldots, \epsilon_n$ are i.i.d. Rademacher random variables, and we refer to the expectation $\Re(\mathcal{G}) = \mathbb{E}_D[\hat{\Re}_D(\mathcal{G})]$ as the Rademacher complexity of $\mathcal{G}$. In addition, we define the worst-case Rademacher complexity as $\tilde{\Re}_n(\mathcal{G}) = \sup_{D \in \mathcal{X}^n} \hat{\Re}_D(\mathcal{G})$, and its expectation is denoted as $\tilde{\Re}(\mathcal{G}) = \mathbb{E}_D[\tilde{\Re}_n(\mathcal{G})]$.*

In multi-label learning, $\boldsymbol{f} \in \mathcal{F}$ is a vector-valued function, which makes traditional Rademacher complexity analysis methods invalid. Hence, we need to convert the Rademacher complexity of a loss function space associated with the vector-valued function class $\mathcal{F}$ into the Rademacher complexity of a tractable scalar-valued function class. The Rademacher complexity can be bounded by other scale-sensitive complexity measures, such as the covering number and fat-shattering dimension [Srebro et al., 2010, Zhang and Zhang, 2023]. The relevant definitions are provided in the appendix.

## 4 General Bounds for LSRL

In this section, we first introduce the assumptions used. Then, we develop a novel vector-contraction inequality for the Rademacher complexity of the loss function space associated with the vector-valued function class $\mathcal{F}$. Finally, with the novel vector-contraction inequality, we derive the generalization bound for general function classes of LSRL with no dependency on the number of labels, up to logarithmic terms, which is tighter than the state of the art. The detailed proofs of the theoretical results in this paper are provided in the appendix.

**Assumption 1.** *Assume that the input features, the loss function and the components of the vector-valued function are bounded: $\|\boldsymbol{x}_i\|_2 \le R$, $\ell(\cdot, \cdot) \le M$, $|f_j(\cdot)| \le B$ for $i \in [n]$, $j \in [c]$ where $R > 0$, $M > 0$ and $B > 0$ are constants.*

**Assumption 2.** *Assume that the loss function $\ell$ is $\rho$-Lipschitz continuous w.r.t. the $\ell_\infty$ norm, that is:*

$$\left|\ell(\boldsymbol{f}(\boldsymbol{x}), \cdot) - \ell(\boldsymbol{f}'(\boldsymbol{x}), \cdot)\right| \le \rho \left\|\boldsymbol{f}(\boldsymbol{x}) - \boldsymbol{f}'(\boldsymbol{x})\right\|_\infty,$$

*where $\rho > 0$, $\|\boldsymbol{t}\|_\infty = \max_{j \in [c]} |t_j|$ for $\boldsymbol{t} = (t_1, \ldots, t_c)$.*

Assumption 1 and 2 are mild assumptions. For Assumption 1, normalization of input features is a common data preprocessing operation. When we consider the function class (1), we often use the assumptions $\|\boldsymbol{w}_j\|_2 \le \Lambda$, $\|\zeta_j(\cdot)\|_2 \le A$ for any $j \in [c]$ to replace the boundedness of the components of the vector-valued function, i.e., $B := \Lambda A$. For Assumption 2, the Lipschitz continuity w.r.t. the $\ell_\infty$ norm has been considered in some literature [Foster and Rakhlin, 2019, Lei et al., 2019, Wu et al., 2021b]. The following Proposition 1 further illustrates that the commonly used loss functions in multi-label learning actually satisfy Assumption 2.

**Proposition 1.** *Assume that the base loss $\ell_b$ defined in Subsection 3.2 is $\mu$-Lipschitz continuous, then the surrogate Hamming Loss is $\mu$-Lipschitz w.r.t. the $\ell_\infty$ norm, the surrogate Subset Loss is $\mu$-Lipschitz w.r.t. the $\ell_\infty$ norm, the surrogate Ranking Loss is $2\mu$-Lipschitz w.r.t. the $\ell_\infty$ norm, and the surrogate Decomposable Loss is $c\mu$-Lipschitz w.r.t. the $\ell_\infty$ norm.*

We define a function class $\mathcal{P}$ consisting of projection operators $p_j : \mathbb{R}^c \mapsto \mathbb{R}$ for any $j \in [c]$ which project the $c$-dimensional vector onto the $j$-th coordinate. Then, we have $\mathcal{P}(\mathcal{F}) = \{(j, \boldsymbol{x}) \mapsto p_j(\boldsymbol{f}(\boldsymbol{x})) : p_j(\boldsymbol{f}(\boldsymbol{x})) = f_j(\boldsymbol{x}), \boldsymbol{f} \in \mathcal{F}, (j, \boldsymbol{x}) \in [c] \times \mathcal{X}\}$. The projection function class decouples the relationship among different components. With the assumption of $\ell_\infty$ norm Lipschitz loss and the above definitions, we show that the Rademacher complexity of the loss function space associated with $\mathcal{F}$ can be bounded by the worst-case Rademacher complexity of the projection function class $\mathcal{P}(\mathcal{F})$. We develop the following novel vector-contraction inequality:

**Lemma 1.** *Let $\mathcal{F}$ be a vector-valued function class of LSRL defined by (1). Let Assumptions 1 and 2 hold. Given a dataset $D$ of size $n$. Then, we have*

$$\hat{\Re}_D(\mathcal{L}) \le 12\sqrt{2}\rho\sqrt{c}\tilde{\Re}_{nc}(\mathcal{P}(\mathcal{F})) \left(1 + \log^{\frac{1}{2}}(8e^2 n^3 c^3) \cdot \log \frac{M\sqrt{n}}{\rho B}\right),$$

*where $\hat{\Re}_D(\mathcal{L}) = \mathbb{E}_{\boldsymbol{\epsilon}} \left[\sup_{\ell \in \mathcal{L}, \boldsymbol{f} \in \mathcal{F}} \frac{1}{n} \sum_{i=1}^n \epsilon_i \ell(\boldsymbol{f}(\boldsymbol{x}_i))\right]$ is the empirical Rademacher complexity of the loss function space associated with $\mathcal{F}$, and $\tilde{\Re}_{nc}(\mathcal{P}(\mathcal{F}))$ is the worst-case Rademacher complexity of the projection function class.*

*Proof Sketch.* First, the Rademacher complexity of the loss function space associated with $\mathcal{F}$ can be bounded by the empirical $\ell_\infty$ norm covering number with the refined Dudley's entropy integral inequality. Second, according to the Lipschitz continuity w.r.t the $\ell_\infty$ norm, the empirical $\ell_\infty$ norm covering number of $\mathcal{F}$ can be bounded by that of $\mathcal{P}(\mathcal{F})$. Third, the empirical $\ell_\infty$ norm covering number of $\mathcal{P}(\mathcal{F})$ can be bounded by the fat-shattering dimension, and the fat-shattering dimension can be bounded by the worst-case Rademacher complexity of $\mathcal{P}(\mathcal{F})$. Hence, the problem is transferred to the estimation of the worst-case Rademacher complexity. Finally, we estimate the lower bound of the worst-case Rademacher complexity of $\mathcal{P}(\mathcal{F})$, and then combined with the above steps, the Rademacher complexity of the loss function space associated with $\mathcal{F}$ can be bounded. $\qquad\square$

With the vector-contraction inequality above, we can derive the following tight bound for LSRL:

**Theorem 1.** *Let $\mathcal{F}$ be a vector-valued function class of LSRL defined by (1). Let Assumptions 1 and 2 hold. Given a dataset $D$ of size $n$. Then, for any $0 < \delta < 1$, with probability at least $1 - \delta$, the following holds for any $\boldsymbol{f} \in \mathcal{F}$:*

$$R(\boldsymbol{f}) \leq \widehat{R}_D(\boldsymbol{f}) + \frac{24\sqrt{2}\rho\Lambda A \left(1 + \log^{\frac{1}{2}}(8e^2 n^3 c^3) \cdot \log\frac{M\sqrt{n}}{\rho B}\right)}{\sqrt{n}} + 3M\sqrt{\frac{\log\frac{2}{\delta}}{2n}}.$$

*Proof Sketch.* We first upper bound the worst-case Rademacher complexity $\tilde{\mathfrak{R}}_{nc}(\mathcal{P}(\mathcal{F}))$, and then combined with Lemma 1, the desired bound can be derived. $\qquad\square$

**Remark 1.** *Although Lemma 1 shows a factor of $\sqrt{c}$, the term $\widetilde{\mathfrak{R}}_{nc}(\mathcal{P}(\mathcal{F})) \leq \Lambda A/\sqrt{nc}$, which makes the Rademacher complexity of the loss function space associated with $\mathcal{F}$ (i.e., $\hat{\mathfrak{R}}_D(\mathcal{L})$) actually independent on $c$, up to logarithmic terms, and results in a tighter bound than the existing $O(c/\sqrt{n})$ and $O(\sqrt{c/n})$ bounds with a faster convergence rate $\widetilde{O}(1/\sqrt{n})$. The bound in Theorem 1 with no dependency on $c$ can provide a general theoretical guarantee for LSRL, even for extreme multi-label learning where the number of labels far exceeds the number of examples [Yu et al., 2014, Prabhu and Varma, 2014, Yen et al., 2016, Liu and Shen, 2019], since it is easy to get that $\log c$ is much smaller than $\sqrt{n}$. The main challenge of generalization analysis for LSRL is that existing theoretical results and existing generalization analysis methods for multi-label learning are not applicable to LSRL. Specifically, existing theoretical bounds often involve the typical vector-contraction inequality [Maurer, 2016] for $\ell_2$ Lipschitz loss: $\mathbb{E}_\epsilon\left[\sup_{\boldsymbol{f}\in\mathcal{F}} \frac{1}{n}\sum_{i=1}^n \epsilon_i \ell(\boldsymbol{f}(\boldsymbol{x}_i))\right] \leq \sqrt{2}\mu\mathbb{E}_\epsilon\left[\sup_{\boldsymbol{f}\in\mathcal{F}} \frac{1}{n}\sum_{i=1}^n\sum_{j=1}^c \epsilon_{ij}f_j(\boldsymbol{x}_i)\right]$, which will lead to bounds with a linear dependency on $c$ for multi-label learning. Lei et al. [2015], Wu and Zhu [2020], Wu et al. [2021a] improve the dependency of the bounds on $c$ to square-root, which preserve the coupling among different components reflected by the constraint $\|\boldsymbol{w}\| \leq \Lambda$. Lei et al. [2019] also improves the bounds of multi-class classification to be independent on $c$ (up to logarithmic terms) for $\ell_\infty$ Lipschitz loss by preserving the coupling among different components, and Wu et al. [2021b] further generalizes these results to multi-label learning. However, LSRL decomposes the multi-label learning problem into $c$ binary classification problems, which means that the relationship among different components needs to be decoupled in the generalization analysis. Hence, how to develop novel vector-contraction inequalities that can induce $\widetilde{O}(1/\sqrt{n})$ bounds and deal with the case of decoupling the relationship among different components are the two most critical difficulties in deriving tighter bounds for LSRL. The introduction of the projection function class plays an important role in solving these two difficulties. It improves the vector-contraction inequalities by a factor of $\sqrt{c}$ and decouples the relationship among different components (which is also reflected by the constraint $\|\boldsymbol{w}_j\| \leq \Lambda$ for any $j \in [c]$). Our tighter $\widetilde{O}(1/\sqrt{n})$ bound in Theorem 1 with no dependency on $c$ (up to logarithmic terms) solve the limitations of existing theoretical results for multi-label learning and can provide a general theoretical guarantee for LSRL.*

The differences in generalization bounds of different LSRL methods are mainly reflected in two aspects. On the one hand, the Lipschitz constant of the loss functions, as we proved in Proposition 1, the Lipschitz constants $\rho$ corresponding to different loss functions are various. On the other hand, the nonlinear mappings induced by different LSRL methods. In fact, when we analyze the generalization for LSRL, we will further have $\|\zeta(\cdot)\| \leq A := \kappa R$ ($\kappa$ is the Lipschitz constant of the nonlinear mappings $\zeta(\cdot)$) to take into account the differences or characteristics of different LSRL methods. We provide detailed analysis for $A$ of typical LSRL methods in the next section.

# 5 Generalization Bounds for Typical LSRL Methods

In this section, we analyze the generalization bounds for several typical LSRL methods, i.e., $k$-means clustering-based [Zhang and Wu, 2015], Lasso-based [Huang et al., 2016] and DNN-based [Hang and Zhang, 2022] LSRL methods. We show that different construction methods of label-specific representation will lead to significant differences in the constant $A$ of the generalization bound in Theorem 1. For each LSRL method, we first give a brief introduction, then give its formal definition corresponding to the class of LSRL defined in (1), and finally derive the generalization bound.

## 5.1 Generalization Bounds for $k$-Means Clustering-Based LSRL Method

As a seminal work, $k$-means clustering-based LSRL method, i.e., LIFT [Zhang and Wu, 2015], uses $k$-means clustering to construct label-specific representation that effectively capture the specific characteristics of each label. Specifically, first, for each label, $k$-means clustering is used to divide the training instances into $K$ clusters, and the centers of the $K$ clusters are obtained, which is denoted as $\boldsymbol{c}_k^j$ for the $j$-th label, $k \in [K]$, $j \in [c]$. Then, these $K$ centers are used to construct the label-specific representation, i.e., in the vector-valued function class of LSRL defined by (1), $\phi_j(\boldsymbol{x}) = \left[ d(\boldsymbol{x}, \boldsymbol{c}_1^j), \ldots, d(\boldsymbol{x}, \boldsymbol{c}_K^j) \right], d(\boldsymbol{x}, \boldsymbol{c}_k^j) = \|\boldsymbol{x} - \boldsymbol{c}_k^j\|$. Finally, a family of $c$ classifiers $f_j$ with $\kappa$-Lipschitz nonlinear mapping are induced based on the generated label-specific representations.

Next, we formally define the process of $k$-means clustering. Here we follow some of the settings and definitions in [Li and Liu, 2021]. Assume that $V : \mathcal{X}^2 \to \mathbb{R}_+$ is a pairwise distance-based function used to measure the dissimilarity between pair observations, and $Z = [Z_1, \ldots, Z_K]$ is a collection of $K$ partition functions $Z_k : \mathcal{X}^2 \to \mathbb{R}_+$ for $k \in [K]$. The clustering framework can be cast as the problem of minimizing the following criterion:

$$\widehat{R}_D(V, Z) = \frac{1}{n(n-1)} \sum_{i,j=1, i \neq j}^{n} \sum_{k=1}^{K} V(\boldsymbol{x}_i, \boldsymbol{x}_j) Z_k(\boldsymbol{x}_i, \boldsymbol{x}_j)$$

over all possible functions $V$ and $Z_k$ for $k \in [K]$. In $k$-means clustering, we have $V(\boldsymbol{x}_i, \boldsymbol{x}_j) = \|\boldsymbol{x}_i - \boldsymbol{x}_j\|_2^2$, and $Z_k(\boldsymbol{x}_i, \boldsymbol{x}_j) = \mathbb{I}\left\{ (\boldsymbol{x}_i, \boldsymbol{x}_j) \in C_k^2 \right\}$, $\boldsymbol{c}_k = \frac{1}{|C_k|} \sum_{\boldsymbol{x} \in C_k} \boldsymbol{x}$, where $C_1, \ldots, C_K$ are the partitions of the feature space $\mathcal{X}$. Let $g_k(\boldsymbol{x}_i, \boldsymbol{x}_j) = V(\boldsymbol{x}_i, \boldsymbol{x}_j) Z_k(\boldsymbol{x}_i, \boldsymbol{x}_j)$ and $\boldsymbol{g} = (g_1, \ldots, g_K)$ be a vector-valued function, $\ell_{clu}(\boldsymbol{g}(\boldsymbol{x}, \boldsymbol{x}')) = \sum_{k=1}^{K} g_k(\boldsymbol{x}, \boldsymbol{x}')$, then $\widehat{R}_D(V, Z)$ can be written as

$$\widehat{R}_D(V, Z) = \frac{1}{n(n-1)} \sum_{i,j=1, i \neq j}^{n} \ell_{clu}(\boldsymbol{g}(\boldsymbol{x}_i, \boldsymbol{x}_j)) = \frac{1}{n(n-1)} \sum_{i,j=1, i \neq j}^{n} \sum_{k=1}^{K} g_k(\boldsymbol{x}_i, \boldsymbol{x}_j).$$

We then define a vector-valued function class of $k$-means clustering as follows:

$$\mathcal{G} = \{ (\boldsymbol{x}, \boldsymbol{x}') \mapsto \boldsymbol{g}(\boldsymbol{x}, \boldsymbol{x}') : \boldsymbol{g}(\boldsymbol{x}, \boldsymbol{x}') = (g_1(\boldsymbol{x}, \boldsymbol{x}'), \ldots, g_K(\boldsymbol{x}, \boldsymbol{x}')),$$
$$g_k(\boldsymbol{x}, \boldsymbol{x}') = \|\boldsymbol{x} - \boldsymbol{x}'\|_2^2 \cdot \mathbb{I}\left\{ (\boldsymbol{x}, \boldsymbol{x}') \in C_k^2 \right\}, \boldsymbol{x}, \boldsymbol{x}' \in \mathcal{X}, k \in [K] \},$$

where $|g_k(\cdot, \cdot)| \leq G$ for $k \in [K]$ and $G > 0$ are constants.

We denote the function class of $k$-means clustering corresponding to the $j$-th label as $\mathcal{G}^j$. With the above definitions, we can derive the tight bound for $k$-means clustering-based LSRL method:

**Theorem 2.** *Let $\mathcal{F}$ be a vector-valued function class of $k$-means clustering-based LSRL defined by (1). Let Assumptions 1 and 2 hold. Given a dataset $D$ of size $n$. Then, for any $0 < \delta < 1$, with probability at least $1 - \delta$, the following holds for any $\boldsymbol{f} \in \mathcal{F}$:*

$$R(\boldsymbol{f}) \leq \widehat{R}_D(\boldsymbol{f}) + 3M \sqrt{\frac{\log \frac{2}{\delta}}{2n}} + \frac{48\sqrt{2}\rho\kappa\Lambda\sqrt{K}R \left( 1 + \log^{\frac{1}{2}}(8e^2n^3c^3) \cdot \log \frac{M\sqrt{n}}{\mu B} \right)}{\sqrt{n}}$$

$$+ \frac{24^2\sqrt{2}\rho\sqrt{K}G}{\sqrt{n}} \left( 1 + \log^{\frac{1}{2}}(e^2n^3c^3) \cdot \log \frac{M\sqrt{nc}}{G} \right) \left( 1 + \log^{\frac{1}{2}}(8e^2n^3c^3) \cdot \log \frac{M\sqrt{n}}{\mu B} \right).$$

**Remark 2.** *There are three key points in the generalization analysis of $k$-means clustering-based LSRL method. 1) Since $k$-means clustering-based LSRL method is two-stage, i.e., the centers of*

*clusters is generated by using $k$-means clustering in the first stage, then these centers are exploited to generate label-specific representations which are used to learn a multi-label classifier in the second stage, we cannot formally express these two stages in a closed-form expression through a composite function ($K$ centers are generated by the $\arg\min$ function). Furthermore, $K$ centers generated in the first stage are actually used as fixed parameters rather than inputs in the second stage. Hence, in order to fully consider the capacity of the model corresponding to the first stage, it is reasonable to define the whole function class as the sum of the function classes $\mathcal{F} + \mathcal{L}_{clu} \circ \mathcal{G}$ corresponding to the methods of these two stages. Then, combined with Lemma 1, the generalization analysis is transformed into the bounding of the complexity of the projection function class $\mathcal{P}(\mathcal{F} + \mathcal{L}_{clu} \circ \mathcal{G})$. The introduction of class $\mathcal{G}$ induces an additional increase in complexity, i.e., the last term in Theorem 2. 2) The generalization analysis of $k$-means clustering-based LSRL method involves the generalization analysis for $k$-means clustering. However, since the $k$-means clustering framework involves pairwise functions, a sequence of pairs of i.i.d. individual observation in $k$-means clustering is no longer independent, which makes standard techniques in the i.i.d case for traditional Rademacher complexity inapplicable for $k$-means clustering. We convert the non-sum-of-i.i.d pairwise function to a sum-of-i.i.d form by using permutations in U-process [Clémençon et al., 2008]. We show that the empirical Rademacher complexity of a loss function space associated with the vector-valued function class $\mathcal{G}$ can be bounded by $\hat{\Re}_{D'}(\mathcal{L}_{clu} \circ \mathcal{G}) := \mathbb{E}_{\boldsymbol{\epsilon}} \left[ \sup_{\boldsymbol{g} \in \mathcal{G}} \frac{1}{\lfloor \frac{n}{2} \rfloor} \sum_{i=1}^{\lfloor \frac{n}{2} \rfloor} \epsilon_i \ell_{clu}(\boldsymbol{g}(\boldsymbol{x}_i, \boldsymbol{x}_{i+\lfloor \frac{n}{2} \rfloor})) \right]$. 3) In order to derive tight bounds for $k$-means clustering, we develop a novel vector-contraction inequality that can induce bounds with a square-root dependency on the number of clusters. The theoretical techniques and results involved here may be of independent interest. The generalization bound of $k$-means clustering-based LSRL method is tighter than the state of the art with a faster convergence rate $\widetilde{O}(\sqrt{K/n})$, which is independent on the number of labels. Since the lower bound for clustering is $\Omega(\sqrt{K/n})$ [Bartlett et al., 1998], our bound is (nearly) optimal, up to logarithmic terms, even from the perspective of clustering. The constant $A$ of the generalization bound in Theorem 1 corresponds to $2\kappa\sqrt{K}R$ here.*

## 5.2 Generalization Bounds for Lasso-Based LSRL Method

Lasso-based LSRL method, i.e., LLSF [Huang et al., 2016], assumes that the label-specific representation of each label should have sparsity and sharing properties. For sparsity, LLSF uses Lasso as the model corresponding to each label. The property of sharing is achieved by considering that two strongly correlated labels will share more features with each other than two uncorrelated or weakly correlated labels and the corresponding weights will be similar, i.e., their inner product will be large.

Formally, since each label corresponds to a Lasso, this means that in the class of LSRL defined by (1), the base loss $\ell_b$ is the squared loss, the nonlinear mappings $\zeta(\cdot)$ and $\phi(\cdot)$ are both identity transformations for any $j \in [c]$, and the constraint $\alpha(\boldsymbol{w})$ is $\|\boldsymbol{w}_j\|_1 \leq \Lambda$ for any $j \in [c]$. For the $j$-th label, the property of sharing is reflected by the additionally introduced constraint $\sum_i^c (1 - s_{ji})\boldsymbol{w}_j^\top \boldsymbol{w}_i \leq \tau$, where $s_{ji}$ is the cosine similarity between labels $y_j$ and $y_i$, here we refer to it as the sharing constraint. The loss function used by Lasso-based LSRL method is the Decomposable loss.

Since the squared loss is not Lipschitz continuous, the theoretical results on the Lipschitz continuity of the loss functions in Proposition 1 cannot be applied to Lasso-based LSRL method. To overcome this challenge, we define the pseudo-Lipschitz function, which is also used in the theoretical analysis of approximate message passing algorithms [Bayati and Montanari, 2011].

**Definition 2.** *For $k \geq 1$, we say that a function $f : \mathbb{R} \to \mathbb{R}$ is pseudo-Lipschitz of order $k$ if there exists a constant $L > 0$ such that the following inequality holds for all $x, y \in \mathbb{R}$:*

$$|f(x) - f(y)| \leq L \left(1 + |x|^{k-1} + |y|^{k-1}\right) |x - y|.$$

Note that any pseudo-Lipschitz function of order 1 is Lipschitz continuous. The following Proposition shows that the Decomposable loss is still Lipschitz continuous if the base loss is the squared loss.

**Proposition 2.** *The squared loss is 1 pseudo-Lipschitz of order 2, the surrogate Decomposable Loss is $(3 + 2B)c$-Lipschitz w.r.t. the $\ell_\infty$ norm if the base loss $\ell_b$ is the squared loss.*

With the above definitions, we can derive the generalization bound for Lasso-based LSRL method:

**Theorem 3.** *Let $\mathcal{F}$ be a vector-valued function class of Lasso-based LSRL defined by (1). Let Assumptions 1 and 2 hold. Given a dataset $D$ of size $n$. Then, for any $0 < \delta < 1$, with probability at*

*least $1 - \delta$, the following holds for any $\boldsymbol{f} \in \mathcal{F}$:*

$$R(\boldsymbol{f}) \leq \widehat{R}_D(\boldsymbol{f}) + \frac{24\sqrt{2}(3 + 2B)c\Lambda R \left(1 + \log^{\frac{1}{2}}(8e^2 n^3 c^3) \cdot \log \frac{M\sqrt{n}}{\rho B}\right)}{\sqrt{n}} + 3M\sqrt{\frac{\log \frac{2}{\delta}}{2n}}.$$

**Remark 3.** *The complexity of the LLSF function class can be bounded by the complexity of the LSRL function class where each label corresponds to a Lasso, since the introduction of the sharing constraint in the LLSF function class reduces the complexity of the function class compared with the LSRL function class where each label corresponds to a Lasso. Hence, the complexity analysis of the LLSF function class can be converted into upper bounding the Rademacher complexity of the LSRL function class where each label corresponds to a Lasso. The constant $A$ of the generalization bound in Theorem 1 corresponds to $R$ here, and the value of $\rho$ is $(3 + 2B)c$, which induce the $\widetilde{O}(c/\sqrt{n})$ bound here. If other loss functions are used, e.g., Hamming, Subset or Ranking loss, instead of Decomposable loss, the dependency of the bounds for Lasso-based LSRL method on $c$ can be improved from linear to independent, up to logarithmic terms.*

## 5.3 Generalization Bounds for DNN-Based LSRL Method

DNN-based LSRL method, i.e., CLIF [Hang and Zhang, 2022], exploits the powerful representation learning capability of deep neural networks (DNNs) to learn label-specific representation in an end-to-end manner. Since the construction of label-specific representation involves graph convolutional networks (GCNs), we first introduce the relevant definitions for GCN.

Let $\mathcal{G} = \{\mathcal{V}, \mathcal{E}\}$ be a given undirected graph, where $\mathcal{V} = \{\boldsymbol{x}_1, \boldsymbol{x}_2, \ldots, \boldsymbol{x}_n\}$ is the set of nodes with size $|\mathcal{V}| = n$ and $\mathcal{E}$ is the set of edges. Let $A$ and $D$ be the adjacency matrix and the diagonal degree matrix respectively, where $D_{ii} = \sum_{j=1}^n A_{ij}$. Let $\tilde{A} = (D + I_n)^{-\frac{1}{2}} (A + I_n) (D + I_n)^{-\frac{1}{2}}$ denote the normalized adjacency matrix with self-connections, where $I$ is the identity matrix. The feature propagation process of a two-layer GCN is $\sigma(\tilde{A}\sigma(\tilde{A}XW_1)W_2)$, where $W_1$ and $W_2$ are parameter matrices, $X$ is the node feature matrix, and the $i$-th row $X_{i*}$ is the node feature $\boldsymbol{x}_i$.

Specifically, for DNN-based LSRL method, first, a graph (here we call it the label graph) is constructed over the label space and a GCN is used to generate the label embeddings, i.e., the label embeddings can be denoted by $\psi(Y) = \sigma_{ReLU}(\tilde{A}\sigma_{ReLU}(\tilde{A}YW_1)W_2)$, where $Y$ is the node feature matrix of the label graph with size $c$ and the nodes are also bounded by $R$, $\sigma_{ReLU}$ is the ReLU activation. Second, the label embedding of the $j$-th label is decoded into the importance vector by a one-layer fully-connected neural network, i.e., $\sigma_{sig}(W_3\psi(Y)_j)$, where $\sigma_{sig}$ is the sigmoid activation, and the input feature is mapped into the latent representation through a one-layer fully-connected neural network, i.e., $\sigma_{ReLU}(W_4\boldsymbol{x})$. Third, for the $j$-th label, the Hadamard product of the importance vector and the latent representation is defined as the pertinent representation, and then the label-specific representation for the $j$-th label is obtained by feeding the pertinent representation into another one-layer fully-connected neural network. Hence, the label-specific representation for the $j$-th label is

$$\phi_j(\boldsymbol{x}) = \sigma_{ReLU} \{W_5 \cdot [\sigma_{ReLU}(W_4\boldsymbol{x}) \odot \sigma_{sig}(W_3\psi(Y)_j)]\}.$$

Finally, the $j$-th model is implemented by a fully-connected layer, i.e., $f_j(\boldsymbol{x}) = \sigma_{sig}(\boldsymbol{w}_j^\top \phi_j(\boldsymbol{x}))$.

In the generalization analysis of the above deep neural network, we introduce the following assumption, which is a common assumption in the generalization analysis for DNNs [Bartlett et al., 2017, Golowich et al., 2018, Zhang and Zhang, 2023, Tang and Liu, 2023].

**Assumption 3.** *Assume that the parameter metrices in DNN-based LSRL method are bounded, i.e., $\|W_i\| \leq D$, $i \in [5]$, where $D > 0$ is a constant.*

With the above definitions, we can derive the generalization bound for DNN-based LSRL method:

**Theorem 4.** *Let $\mathcal{F}$ be a vector-valued function class of DNN-based LSRL defined by (1). Let Assumptions 1, 2, and 3 hold. Given a dataset $D$ of size $n$. Then, for any $0 < \delta < 1$, with probability at least $1 - \delta$, the following holds for any $\boldsymbol{f} \in \mathcal{F}$:*

$$R(\boldsymbol{f}) \leq \widehat{R}_D(\boldsymbol{f}) + \frac{6\sqrt{2}\rho\Lambda D^5 R^2(g_{\max} + 1) \left(1 + \log^{\frac{1}{2}}(8e^2 n^3 c^3) \cdot \log \frac{M\sqrt{n}}{\rho B}\right)}{\sqrt{n}} + 3M\sqrt{\frac{\log \frac{2}{\delta}}{2n}},$$

*where $g_{\max}$ is the maximum node degree of the label graph.*

**Remark 4.** *The term $\widetilde{\Re}_{nc}(\mathcal{P}(\mathcal{F})) \leq \Lambda D^5 R^2 (g_{\max} + 1)/4\sqrt{nc}$, which makes the Rademacher complexity $\mathring{\Re}_D(\mathcal{L})$ actually independent on $c$, up to logarithmic terms, and results in a tight bound for DNN-based LSRL method with a faster convergence rate $\widetilde{O}(1/\sqrt{n})$. The constant $A$ of the generalization bound in Theorem 1 corresponds to $D^5 R^2 (g_{\max} + 1)/4$ here. For deep GCNs, the increase in depth means that the generalization performance will deteriorate, which is consistent with empirical performance and guides us not to design too many layers of GCN. In addition, the bound is linearly dependent on the maximum node degree of the label graph, which suggests that when the performance of the model is always unsatisfactory, we can check whether the maximum node degree is large and consider using some techniques to remove some edges, e.g., DropEdge [Rong et al., 2020], to alleviate the over-fitting problem. We will further explore more network structures to learn more effective label-specific representations, e.g., hypernetworks [Galanti and Wolf, 2020, Chen et al., 2023, Shen et al., 2023], deep kernel networks [Zhang and Liao, 2020, Zhang and Zhang, 2023], and provide generalization analysis for the corresponding DNN-based LSRL methods.*

## 6   Discussion

Compared with existing methods considering label correlations, which mainly focus on the processing of the label space by exploiting or modeling relationships between labels, LSRL mainly focuses on the operation on the input space and implicitly considers label correlations in the process of constructing label-specific representations. For example, in the construction of label-specific representations in LLSF and CLIF, the label correlation information is embedded into the label-specific representations in the input space by introducing the sharing constraint and using a GCN over the label graph to generate label embeddings, respectively. LSRL methods are more effective since the label correlation information is considered in the construction of label-specific representations.

Our theoretical results explain why LSRL is an effective strategy to improve the generalization performance of multi-label learning. On the one hand, existing results can improve the dependency of the bound on $c$ from linear to square-root by preserving the coupling among different components, which corresponds to high-order label correlations induced by norm regularizers. However, the improvement in the preservation of coupling by a factor of $\sqrt{c}$ benefits from replacing $\sqrt{c}\Lambda$ with $\Lambda$ in the constraint to some extent, and preserving the coupling corresponds to the stricter assumption [Zhang and Zhang, 2024]. Our results for LSRL decouple the relationship among different components, and the bounds with a weaker dependency on $c$ are tighter than the existing results that preserve the coupling, which also explains why LSRL methods outperform the multi-label methods that consider high-order label correlations induced by norm regularizers. On the other hand, based on our results, we can find that LSRL methods substantially increase the data processing, i.e., the process of constructing label-specific representations. From the perspective of model capacity, compared with traditional multi-label methods, since the introduction of construction methods of label-specific representations, the capacity of the model is significantly increased, especially if deep learning methods are used to generate label-specific representations, which improves the representation ability of the model. Or more intuitively, LSRL means an increase in model capacity and stronger representation ability, which makes it easier to find the hypotheses with better generalization in the function class.

The vector-contraction inequality and the theoretical tools developed here are applicable to the theoretical analysis of other problem settings, such as multi-class classification, or more general vector-valued learning problem. For multi-class classification, multi-class margin-based loss, multinomial logistic loss, Top-$k$ hinge loss, etc. are all $\ell_\infty$ Lipschitz [Lei et al., 2019]. For multi-label learning, the surrogate loss for Macro-Averaged AUC is also $\ell_\infty$ Lipschitz [Zhang and Zhang, 2024].

## 7   Conclusion

In this paper, we propose a novel vector-contraction inequality for $\ell_\infty$ norm Lipschitz continuous loss, and derive bounds for general function classes of LSRL with a weaker dependency on $c$ than the state of the art. In addition, we analyze the bounds for several typical LSRL methods, and study the impact of different label-specific representations on the generalization analysis.

In future work, we will extend our bounds to more LSRL methods, and derive tighter bounds for LSRL with a faster convergence rate w.r.t. the number of examples, and further design efficient models and algorithms to construct label-specific representations with good generalization performance.

## Acknowledgements

The authors wish to thank the anonymous reviewers for their helpful comments and suggestions. This work was supported by the National Science Foundation of China (62225602) and the Big Data Computing Center of Southeast University.

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

# A  Appendix

## A.1  Appendix Outline

In the appendix, we give the detailed proofs of those theoretical results in the main paper. Our main proofs include:

- The $\ell_\infty$ Lipschitz continuity of the commonly used losses for LSRL (Proposition 1).
- The novel vector-contraction inequality for $\ell_\infty$ Lipschitz loss (Lemma 1).
- The generalization bound of the general LSRL class with no dependency on $c$ (Theorem 1).
- The generalization bound for $k$-means clustering-based LSRL method (Theorem 2).
- The (pseudo-) Lipschitz continuity of the squared and Decomposable loss (Proposition 2).
- The generalization bound for Lasso-based LSRL method (Theorem 3).
- The generalization bound for DNN-based LSRL method (Theorem 4).

## A.2  Preliminaries

### A.2.1  Definitions of the corresponding complexity measures

**Definition 3** ($\ell_\infty$ norm covering number). *Let $\mathcal{F}$ be a class of real-valued functions mapping from $\mathcal{X}$ to $\mathbb{R}$. Let $D = \{\boldsymbol{x}_1, \ldots, \boldsymbol{x}_n\}$ be a set with $n$ i.i.d. samples. For any $\epsilon > 0$, the empirical $\ell_\infty$ **norm covering number** $\mathcal{N}_\infty(\epsilon, \mathcal{F}, D)$ w.r.t. $D$ is defined as the minimal number $m$ of a collection of vectors $\boldsymbol{v}^1, \ldots, \boldsymbol{v}^m \in \mathbb{R}^n$ such that $\max_{i \in [n]} \left| f(\boldsymbol{x}_i) - \boldsymbol{v}_i^j \right| \le \epsilon$ ($\boldsymbol{v}_i^j$ is the $i$-th component of the vector $\boldsymbol{v}^j$). In this case, we call $\{\boldsymbol{v}^1, \ldots, \boldsymbol{v}^m\}$ an $(\epsilon, \ell_\infty)$-cover of $\mathcal{F}$ with respect to $D$. We also define $\mathcal{N}_\infty(\epsilon, \mathcal{F}, n) = \sup_D \mathcal{N}_\infty(\epsilon, \mathcal{F}, D)$.*

**Definition 4** (Fat-shattering dimension). *Let $\mathcal{F}$ be a class of real-valued functions mapping from $\mathcal{X}$ to $\mathbb{R}$. We define the fat-shattering dimension $\mathrm{fat}_\epsilon(\mathcal{F})$ at scale $\epsilon > 0$ as the largest $p \in \mathbb{N}$ such that there exist $p$ points $\boldsymbol{x}_1, \ldots, \boldsymbol{x}_p \in \mathcal{X}$ and witnesses $s_1, \ldots, s_p \in \mathbb{R}$ satisfying: for any $\delta_1, \ldots, \delta_p \in \{-1, +1\}$ there exists $f \in \mathcal{F}$ with $\delta_i (f(\boldsymbol{x}_i) - s_i) \ge \epsilon, \forall i = 1, \ldots, p$.*

### A.2.2  The Bound for the loss function space

For any training dataset $D = \{(\boldsymbol{x}_i, \boldsymbol{y}_i) : i \in [n]\}$, let $D' = \{(\boldsymbol{x}_i, \boldsymbol{y}_i) : i \in [n]\}$ be the training dataset with only one sample different from $D$, where the $k$-th sample is replaced by $(\boldsymbol{x}_k', \boldsymbol{y}_k')$. Let $\Phi(D) = \sup_{\boldsymbol{f} \in \mathcal{F}} [\mathbb{E}_{(\boldsymbol{x}, \boldsymbol{y}) \sim P}[\ell(\boldsymbol{f}(\boldsymbol{x}), \boldsymbol{y})] - \frac{1}{n} \sum_{i=1}^n \ell(\boldsymbol{f}(\boldsymbol{x}_i), \boldsymbol{y}_i)] = \sup_{\boldsymbol{f} \in \mathcal{F}} [R(\boldsymbol{f}) - \widehat{R}_D(\boldsymbol{f})]$, then

$$
\begin{aligned}
&\Phi(D') - \Phi(D) \\
&= \sup_{\boldsymbol{f} \in \mathcal{F}} [R(\boldsymbol{f}) - \widehat{R}_{D'}(\boldsymbol{f})] - \sup_{\boldsymbol{f} \in \mathcal{F}} [R(\boldsymbol{f}) - \widehat{R}_D(\boldsymbol{f})] \\
&\le \sup_{\boldsymbol{f} \in \mathcal{F}} [\widehat{R}_D(\boldsymbol{f}) - \widehat{R}_{D'}(\boldsymbol{f})] \\
&= \sup_{\boldsymbol{f} \in \mathcal{F}} \frac{[\ell(\boldsymbol{f}(\boldsymbol{x}_k), \boldsymbol{y}_k) - \ell(\boldsymbol{f}(\boldsymbol{x}_k'), \boldsymbol{y}_k')]}{n} \\
&\le \frac{M}{n}.
\end{aligned}
$$

According to McDiarmid's inequality, for any $0 < \delta < 1$, with probability at least $1 - \frac{\delta}{2}$ over the training dataset $D$, the following holds:

$$
\Phi(D) \le \mathbb{E}_D[\Phi(D)] + M \sqrt{\frac{\ln(2/\delta)}{2n}}. \tag{2}
$$

Then, we will estimate the upper bound of $\mathbb{E}_D[\Phi(D)]$.

$$\mathbb{E}_D[\Phi(D)]$$

$$=\mathbb{E}_D\left[\sup_{\boldsymbol{f}\in\mathcal{F}}\left[\mathbb{E}_{D'}\left[\widehat{R}_{D'}(\boldsymbol{f})-\widehat{R}_D(\boldsymbol{f})\right]\right]\right]$$

$$\leq\mathbb{E}_{D,D'}\left[\sup_{\boldsymbol{f}\in\mathcal{F}}\left[\widehat{R}_{D'}(\boldsymbol{f})-\widehat{R}_D(\boldsymbol{f})\right]\right]$$

$$=\mathbb{E}_{D,D'}\left[\sup_{\boldsymbol{f}\in\mathcal{F}}\frac{1}{n}\left[\sum_{i=1}^{n}\ell(\boldsymbol{f}(\boldsymbol{x}_i'),\boldsymbol{y}_i')-\ell(\boldsymbol{f}(\boldsymbol{x}_i),\boldsymbol{y}_i)\right]\right]$$

$$=\mathbb{E}_{\boldsymbol{\epsilon},D,D'}\left[\sup_{\boldsymbol{f}\in\mathcal{F}}\frac{1}{n}\left[\sum_{i=1}^{n}\epsilon_i\left(\ell(\boldsymbol{f}(\boldsymbol{x}_i'),\boldsymbol{y}_i'))-\ell(\boldsymbol{f}(\boldsymbol{x}_i),\boldsymbol{y}_i)\right)\right]\right]$$

$$\leq\mathbb{E}_{\boldsymbol{\epsilon},D'}\left[\sup_{\boldsymbol{f}\in\mathcal{F}}\frac{1}{n}\sum_{i=1}^{n}\epsilon_i\ell(\boldsymbol{f}(\boldsymbol{x}_i'),\boldsymbol{y}_i')\right]+\mathbb{E}_{\boldsymbol{\epsilon},D}\left[\sup_{\boldsymbol{f}\in\mathcal{F}}\frac{1}{n}\sum_{i=1}^{n}-\epsilon_i\ell(\boldsymbol{f}(\boldsymbol{x}_i),\boldsymbol{y}_i)\right]$$

$$=2\mathbb{E}_{\boldsymbol{\epsilon},D}\left[\sup_{\boldsymbol{f}\in\mathcal{F}}\frac{1}{n}\sum_{i=1}^{n}\epsilon_i\ell(\boldsymbol{f}(\boldsymbol{x}_i),\boldsymbol{y}_i)\right]. \tag{3}$$

Then apply McDiarmid's inequality to $\mathbb{E}_{\boldsymbol{\epsilon}}\left[\sup_{\boldsymbol{f}\in\mathcal{F}}\frac{1}{n}\sum_{i=1}^{n}\epsilon_i\ell(\boldsymbol{f}(\boldsymbol{x}_i),\boldsymbol{y}_i)\right]$, we have

$$\mathbb{E}_{\boldsymbol{\epsilon},D}\left[\sup_{\boldsymbol{f}\in\mathcal{F}}\frac{1}{n}\sum_{i=1}^{n}\epsilon_i\ell(\boldsymbol{f}(\boldsymbol{x}_i),\boldsymbol{y}_i)\right]\leq\mathbb{E}_{\boldsymbol{\epsilon}}\left[\sup_{\boldsymbol{f}\in\mathcal{F}}\frac{1}{n}\sum_{i=1}^{n}\epsilon_i\ell(\boldsymbol{f}(\boldsymbol{x}_i),\boldsymbol{y}_i)\right]+M\sqrt{\frac{\ln(2/\delta)}{2n}},$$

i.e.,

$$\Re(\mathcal{L})\leq\hat{\Re}_D(\mathcal{L})+M\sqrt{\frac{\ln(2/\delta)}{2n}}. \tag{4}$$

Combining with (2), (3) and (4), then

$$R(\boldsymbol{f})\leq\widehat{R}_D(\boldsymbol{f})+2\hat{\Re}_D(\mathcal{L})+3M\sqrt{\frac{\log\frac{2}{\delta}}{2n}}. \tag{5}$$

## A.3 General Bounds for LSRL

### A.3.1 Proof of Proposition 1

We first prove that the surrogate Hamming Loss is $\mu$-Lipschitz continuous with respect to the $\ell_\infty$ norm.

$$\left|\ell_H(\boldsymbol{f}(\boldsymbol{x}),\boldsymbol{y})-\ell_H\left(\boldsymbol{f}'(\boldsymbol{x}),\boldsymbol{y}\right)\right|$$

$$=\left|\frac{1}{c}\sum_{j=1}^{c}\ell_b\left(y_jf_j(\boldsymbol{x})\right)-\frac{1}{c}\sum_{j=1}^{c}\ell_b\left(y_jf_j'(\boldsymbol{x})\right)\right|$$

$$=\frac{1}{c}\sum_{j=1}^{c}\left|\ell_b\left(y_jf_j(\boldsymbol{x})\right)-\ell_b\left(y_jf_j'(\boldsymbol{x})\right)\right|$$

$$\leq\frac{1}{c}\sum_{j=1}^{c}\mu\left|f_j(\boldsymbol{x})-f_j'(\boldsymbol{x})\right|$$

$$\leq\frac{1}{c}\mu c\max_{j\in[c]}\left|f_j(\boldsymbol{x})-f_j'(\boldsymbol{x})\right|$$

$$=\mu\left\|\boldsymbol{f}(\boldsymbol{x})-\boldsymbol{f}'(\boldsymbol{x})\right\|_\infty.$$

Second, with the elementary inequality

$$\left|\max\left\{a_1,\ldots,a_c\right\} - \max\left\{b_1,\ldots,b_c\right\}\right| \leq \max\left\{\left|a_1 - b_1\right|,\ldots,\left|a_c - b_c\right|\right\},$$

we prove that the surrogate Subset Loss is $\mu$-Lipschitz continuous with respect to the $\ell_\infty$ norm.

$$\left|\ell_S(\boldsymbol{f}(\boldsymbol{x}),\boldsymbol{y}) - \ell_S\left(\boldsymbol{f}'(\boldsymbol{x}),\boldsymbol{y}\right)\right|$$

$$= \left|\max_{j\in[c]} \ell_b\left(y_j f_j(\boldsymbol{x})\right) - \max_{j\in[c]} \ell_b\left(y_j f_j'(\boldsymbol{x})\right)\right|$$

$$\leq \max_{j\in[c]} \left|\ell_b\left(y_j f_j(\boldsymbol{x})\right) - \ell_b\left(y_j f_j'(\boldsymbol{x})\right)\right|$$

$$\leq \mu \max_{j\in[c]} \left|f_j(\boldsymbol{x}) - f_j'(\boldsymbol{x})\right|$$

$$= \mu \left\|\boldsymbol{f}(\boldsymbol{x}) - \boldsymbol{f}'(\boldsymbol{x})\right\|_\infty.$$

Third, we prove that the surrogate Ranking Loss is $2\mu$-Lipschitz continuous with respect to the $\ell_\infty$ norm.

$$\left|\ell_R(\boldsymbol{f}(\boldsymbol{x}),\boldsymbol{y}) - \ell_R\left(\boldsymbol{f}'(\boldsymbol{x}),\boldsymbol{y}\right)\right|$$

$$= \frac{1}{|Y^+||Y^-|}\left|\sum_{p\in Y^+}\sum_{q\in Y^-}\left(\ell_b\left(f_p(\boldsymbol{x}) - f_q(\boldsymbol{x})\right) - \ell_b\left(f_p'(\boldsymbol{x}) - f_q'(\boldsymbol{x})\right)\right)\right|$$

$$\leq \max_{p\in Y^+,q\in Y^-}\left|\ell_b\left(f_p(\boldsymbol{x}) - f_q(\boldsymbol{x})\right) - \ell_b\left(f_p'(\boldsymbol{x}) - f_q'(\boldsymbol{x})\right)\right|$$

$$\leq \mu \max_{p\in Y^+,q\in Y^-}\left|\left(f_p(\boldsymbol{x}) - f_q(\boldsymbol{x})\right) - \left(f_p'(\boldsymbol{x}) - f_q'(\boldsymbol{x})\right)\right|$$

$$\leq \mu\left(\max_{p\in Y^+}\left|f_p(\boldsymbol{x}) - f_p'(\boldsymbol{x})\right| + \max_{q\in Y^-}\left|f_q(\boldsymbol{x}) - f_q'(\boldsymbol{x})\right|\right)$$

$$\leq 2\mu \max_{j\in[c]}\left|f_j(\boldsymbol{x}) - f_j'(\boldsymbol{x})\right|$$

$$= 2\mu\left\|\boldsymbol{f}(\boldsymbol{x}) - \boldsymbol{f}'(\boldsymbol{x})\right\|_\infty.$$

Finally, we prove that the surrogate Decomposable Loss is $c\mu$-Lipschitz continuous with respect to the $\ell_\infty$ norm.

$$\left|\ell_D(\boldsymbol{f}(\boldsymbol{x}),\boldsymbol{y}) - \ell_D\left(\boldsymbol{f}'(\boldsymbol{x}),\boldsymbol{y}\right)\right|$$

$$= \left|\sum_{j=1}^{c} \ell_b\left(y_j f_j(\boldsymbol{x})\right) - \sum_{j=1}^{c} \ell_b\left(y_j f_j'(\boldsymbol{x})\right)\right|$$

$$= \sum_{j=1}^{c}\left|\ell_b\left(y_j f_j(\boldsymbol{x})\right) - \ell_b\left(y_j f_j'(\boldsymbol{x})\right)\right|$$

$$\leq \sum_{j=1}^{c} \mu\left|f_j(\boldsymbol{x}) - f_j'(\boldsymbol{x})\right|$$

$$\leq c\mu \max_{j\in[c]}\left|f_j(\boldsymbol{x}) - f_j'(\boldsymbol{x})\right|$$

$$= c\mu\left\|\boldsymbol{f}(\boldsymbol{x}) - \boldsymbol{f}'(\boldsymbol{x})\right\|_\infty.$$

### A.3.2 Proof of Lemma 1

**Proof Sketch**: First, the Rademacher complexity of the loss function space associated with $\mathcal{F}$ can be bounded by the empirical $\ell_\infty$ norm covering number with the refined Dudley's entropy integral inequality. Second, according to the Lipschitz continuity w.r.t the $\ell_\infty$ norm, the empirical $\ell_\infty$ norm covering number of $\mathcal{F}$ can be bounded by the empirical $\ell_\infty$ norm covering number of $\mathcal{P}(\mathcal{F})$. Third, the empirical $\ell_\infty$ norm covering number of $\mathcal{P}(\mathcal{F})$ can be bounded by the fat-shattering dimension, and the fat-shattering dimension can be bounded by the worst-case Rademacher complexity of $\mathcal{P}(\mathcal{F})$.

Hence, the problem is transferred to the estimation of the worst-case Rademacher complexity. Finally, we estimate the lower bound of the worst-case Rademacher complexity of $\mathcal{P}(\mathcal{F})$, and then combined with the above steps, the Rademacher complexity of the loss function space associated with $\mathcal{F}$ can be bounded.

We first introduce the following lemmas:

**Lemma 2** (Khintchine-Kahane inequality [Lust-Piquard and Pisier, 1991]). *Let $\boldsymbol{v}_1, \ldots, \boldsymbol{v}_n \in \mathcal{H}$, where $\mathcal{H}$ is a Hilbert space with $\|\cdot\|$ being the associated p-th norm. Let $\epsilon_1, \ldots, \epsilon_n$ be a sequence of independent Rademacher variables. Then, for any $p \geq 1$ there holds*

$$\min(\sqrt{p-1}, 1) \left[ \sum_{i=1}^n \|\boldsymbol{v}_i\|^2 \right]^{\frac{1}{2}} \leq \left[ \mathbb{E}_{\boldsymbol{\epsilon}} \left\| \sum_{i=1}^n \epsilon_i \boldsymbol{v}_i \right\|^p \right]^{\frac{1}{p}} \leq \max(\sqrt{p-1}, 1) \left[ \sum_{i=1}^n \|\boldsymbol{v}_i\|^2 \right]^{\frac{1}{2}},$$

*and*

$$\mathbb{E}_{\boldsymbol{\epsilon}} \left\| \sum_{i=1}^n \epsilon_i \boldsymbol{v}_i \right\| \geq 2^{-\frac{1}{2}} \left[ \sum_{i=1}^n \|\boldsymbol{v}_i\|^2 \right]^{\frac{1}{2}}.$$

**Lemma 3** (Lemma A.2 in [Srebro et al., 2010]). *For any function class $\mathcal{F}$, any $S$ with a finite sample of size $n$ and any $\epsilon > \hat{\mathfrak{R}}_S(\mathcal{F})$, we have that*

$$\text{fat}_\epsilon(\mathcal{F}) \leq \frac{4n\hat{\mathfrak{R}}_S^2(\mathcal{F})}{\epsilon^2}.$$

**Lemma 4** ([Srebro et al., 2010, Lei et al., 2019]). *If any function in class $\mathcal{F}$ takes values in $[-B, B]$, then for any $S$ with a finite sample of size $n$, any $\epsilon > 0$ with $\text{fat}_\epsilon(\mathcal{F}) < n$, we have*

$$\log \mathcal{N}_\infty (\epsilon, \mathcal{F}, S) \leq \text{fat}_\epsilon(\mathcal{F}) \log \frac{2eBn}{\epsilon}.$$

The following lemma is a refined result of Proposition 22 in [Ledent et al., 2021], where we replace the function class taking values in $[0, 1]$ with the $B$-bounded function class, the refinement is obvious.

**Lemma 5** (Refined Dudley's entropy integral inequality). *Let $\mathcal{F}$ be a real-valued function class with $f \leq B$, $f \in \mathcal{F}$, $B > 0$, and assume that $0 \in \mathcal{F}$. Let $S$ be a finite sample of size $n$. For any $2 \leq p \leq \infty$, we have the following relationship between the Rademacher complexity $\hat{\mathfrak{R}}_S(\mathcal{F})$ and the covering number $\mathcal{N}_p(\epsilon, \mathcal{F}, S)$.*

$$\hat{\mathfrak{R}}_S(\mathcal{F}) \leq \inf_{\alpha > 0} \left( 4\alpha + \frac{12}{\sqrt{n}} \int_\alpha^B \sqrt{\log \mathcal{N}_p(\epsilon, \mathcal{F}, S)} d\epsilon \right).$$

**Step 1**: We first derive the relationship between the empirical $\ell_\infty$ norm covering number $\mathcal{N}_\infty(\epsilon, \mathcal{L}, D)$ and the empirical $\ell_\infty$ norm covering number $\mathcal{N}_\infty(\epsilon, \mathcal{P}(\mathcal{F}), [c] \times D)$.

For the dataset $D = \{(\boldsymbol{x}_1, \boldsymbol{y}_1), \ldots, (\boldsymbol{x}_n, \boldsymbol{y}_n)\}$ with $n$ i.i.d. examples:

$$\max_i |\ell(\boldsymbol{f}(\boldsymbol{x}_i), \boldsymbol{y}_i) - \ell(\boldsymbol{f}'(\boldsymbol{x}_i), \boldsymbol{y}_i)|$$

$$\leq \rho \max_i \|\boldsymbol{f}(\boldsymbol{x}_i) - \boldsymbol{f}'(\boldsymbol{x}_i)\|_\infty \quad \text{(Use Assumption 2)}$$

$$\leq \rho \max_i \max_j |f_j(\boldsymbol{x}_i) - f_j'(\boldsymbol{x}_i)|$$

$$\leq \rho \max_i \max_j |p_j(\boldsymbol{f}(\boldsymbol{x}_i)) - p_j(\boldsymbol{f}'(\boldsymbol{x}_i))|. \quad \text{(The definition of the projection function class } \mathcal{P}(\mathcal{F}))$$

Then, according to the definition of the empirical $\ell_\infty$ covering number, we have that an empirical $\ell_\infty$ cover of $\mathcal{P}(\mathcal{F})$ at radius $\epsilon/\rho$ is also an empirical $\ell_\infty$ cover of the loss function space associated with $\mathcal{F}$ at radius $\epsilon$, and we can conclude that:

$$\mathcal{N}_\infty (\epsilon, \mathcal{L}, D) \leq \mathcal{N}_\infty \left( \frac{\epsilon}{\rho}, \mathcal{P}(\mathcal{F}), [c] \times D \right). \tag{6}$$

**Step 2**: We show that the empirical $\ell_\infty$ norm covering number of $\mathcal{P}(\mathcal{F})$ can be bounded by the fat-shattering dimension, and the fat-shattering dimension can be bounded by the worst-case Rademacher complexity of $\mathcal{P}(\mathcal{F})$.

According to Lemma 3, for any $\epsilon > \hat{\mathfrak{R}}_{[c] \times D}(\mathcal{P}(\mathcal{F}))$, we have

$$\text{fat}_\epsilon(\mathcal{P}(\mathcal{F})) \leq \frac{4nc\hat{\mathfrak{R}}^2_{[c] \times D}(\mathcal{P}(\mathcal{F}))}{\epsilon^2}.$$

Then, combining with Lemma 4, we have

$$\log \mathcal{N}_\infty\left(\epsilon, \mathcal{P}(\mathcal{F}), [c] \times D\right) \leq \frac{4nc\hat{\mathfrak{R}}^2_{[c] \times D}(\mathcal{P}(\mathcal{F}))}{\epsilon^2} \log \frac{2eBnc}{\epsilon}.$$

Use inequality $\epsilon > \hat{\mathfrak{R}}_{[c] \times D}(\mathcal{P}(\mathcal{F}))$, we have

$$\log \mathcal{N}_\infty(\epsilon, \mathcal{P}(\mathcal{F}), [c] \times D) \leq \frac{4nc\hat{\mathfrak{R}}^2_{[c] \times D}(\mathcal{P}(\mathcal{F}))}{\epsilon^2} \log \frac{2eBnc}{\hat{\mathfrak{R}}_{[c] \times D}(\mathcal{P}(\mathcal{F}))}. \tag{7}$$

**Step 3**: According to Assumption 1 in the main paper, we can obtain the lower bound of the worst-case Rademacher complexity $\widetilde{\mathfrak{R}}_{nc}(\mathcal{P}(\mathcal{F}))$ by the Khintchine-Kahane inequality with $p = 1$:

$$
\begin{aligned}
&\widetilde{\mathfrak{R}}_{nc}(\mathcal{P}(\mathcal{F})) \\
&= \sup_{[c] \times D \in [c] \times \mathcal{X}^n} \hat{\mathfrak{R}}_{[c] \times D}(\mathcal{P}(\mathcal{F})) \\
&= \sup_{[c] \times D \in [c] \times \mathcal{X}^n} \mathbb{E}_{\boldsymbol{\epsilon}} \left[ \sup_{p_j(\boldsymbol{f}(\boldsymbol{x}_i)) \in \mathcal{P}(\mathcal{F})} \frac{1}{nc} \sum_{i=1}^n \sum_{j=1}^c \epsilon_{ij} p_j(\boldsymbol{f}(\boldsymbol{x}_i)) \right] \\
&= \sup_{[c] \times D \in [c] \times \mathcal{X}^n} \mathbb{E}_{\boldsymbol{\epsilon}} \left[ \sup_{f_j \in \mathcal{F}_j} \frac{1}{nc} \sum_{i=1}^n \sum_{j=1}^c \epsilon_{ij} f_j(\boldsymbol{x}_i) \right] \\
&= \sup_{\|\zeta_j(\phi_j(\boldsymbol{x}_i))\|_2 \leq A : i \in [n], j \in [c]} \frac{1}{nc} \mathbb{E}_{\boldsymbol{\epsilon}} \left[ \sup_{\|\boldsymbol{w}_j\|_2 \leq \Lambda} \sum_{i=1}^n \sum_{j=1}^c \epsilon_{ij} \langle \boldsymbol{w}_j, \zeta_j(\phi_j(\boldsymbol{x}_i)) \rangle \right] \\
&= \sup_{\|\zeta_j(\phi_j(\boldsymbol{x}_i))\|_2 \leq A : i \in [n], j \in [c]} \frac{\Lambda}{nc} \mathbb{E}_{\boldsymbol{\epsilon}} \| \sum_{i=1}^n \sum_{j=1}^c \epsilon_{ij} \zeta_j(\phi_j(\boldsymbol{x}_i)) \| \\
&\geq \sup_{\|\zeta_j(\phi_j(\boldsymbol{x}_i))\|_2 \leq A : i \in [n], j \in [c]} \frac{\Lambda}{nc} \frac{1}{\sqrt{2}} \left[ \sum_{i=1}^n \sum_{j=1}^c \|\zeta_j(\phi_j(\boldsymbol{x}_i))\|^2 \right]^{\frac{1}{2}}. \quad \text{(Use Lemma 2)}
\end{aligned}
$$

Since $\|\zeta_j(\phi_j(\boldsymbol{x}_i))\|_2 \leq A$, we set $\sup_{\|\zeta_j(\phi_j(\boldsymbol{x}_i))\|_2 \leq A : i \in [n], j \in [c]} \frac{1}{nc} \left[ \sum_{i=1}^n \sum_{j=1}^c \|\zeta_j(\phi_j(\boldsymbol{x}_i))\|^2 \right]^{\frac{1}{2}} = \frac{A}{\sqrt{nc}}$. So,

$$\widetilde{\mathfrak{R}}_{nc}(\mathcal{P}(\mathcal{F})) \geq \frac{\Lambda A}{\sqrt{2nc}} = \frac{B}{\sqrt{2nc}}. \tag{8}$$

**Step 4**: According to Lemma 5 and combined with the above steps, we have

$$\hat{\Re}_D(\mathcal{L})$$

$$\leq \inf_{\alpha>0}\left(4\alpha + \frac{12}{\sqrt{n}}\int_\alpha^M \sqrt{\log\mathcal{N}_\infty(\epsilon,\mathcal{L},D)}d\epsilon\right)$$

$$\leq \inf_{\alpha>0}\left(4\alpha + \frac{12}{\sqrt{n}}\int_\alpha^M \sqrt{\log\mathcal{N}_\infty(\frac{\epsilon}{\rho},\mathcal{P}(\mathcal{F}),[c]\times D)}d\epsilon\right) \quad \text{(Use inequality (6))}$$

$$\leq \inf_{\alpha>0}\left(4\alpha + \frac{12}{\sqrt{n}}\int_\alpha^M \sqrt{\frac{4nc\rho^2\hat{\Re}^2_{[c]\times D}(\mathcal{P}(\mathcal{F}))}{\epsilon^2}\log\frac{2eBnc}{\hat{\Re}_{[c]\times D}(\mathcal{P}(\mathcal{F}))}}d\epsilon\right) \quad \text{(Use inequality (7))}$$

$$\leq \inf_{\alpha>0}\left(4\alpha + \frac{12}{\sqrt{n}}\int_\alpha^M \sqrt{\frac{4nc\rho^2\widetilde{\Re}^2_{nc}(\mathcal{P}(\mathcal{F}))}{\epsilon^2}\log(2\sqrt{2}eBn^{\frac{3}{2}}c^{\frac{3}{2}})}d\epsilon\right)$$

(Use inequality (8) and the definition of the worst-case Rademacher complexity)

$$\leq \inf_{\alpha>0}\left(4\alpha + 12\sqrt{2}\rho\sqrt{c}\widetilde{\Re}_{nc}(\mathcal{P}(\mathcal{F}))\log^{\frac{1}{2}}(8e^2n^3c^3)\int_\alpha^M \epsilon^{-1}d\epsilon\right)$$

$$\leq 12\sqrt{2}\rho\sqrt{c}\widetilde{\Re}_{nc}(\mathcal{P}(\mathcal{F})) + 12\sqrt{2}\rho\sqrt{c}\widetilde{\Re}_{nc}(\mathcal{P}(\mathcal{F}))\log^{\frac{1}{2}}(8e^2n^3c^3)\cdot\log\frac{M}{3\sqrt{2}\rho\sqrt{c}\widetilde{\Re}_{nc}(\mathcal{P}(\mathcal{F}))}$$

(Choose $\alpha = 3\sqrt{2}\rho\sqrt{c}\widetilde{\Re}_{nc}(\mathcal{P}(\mathcal{F}))$)

$$\leq 12\sqrt{2}\rho\sqrt{c}\widetilde{\Re}_{nc}(\mathcal{P}(\mathcal{F})) + 12\sqrt{2}\rho\sqrt{c}\widetilde{\Re}_{nc}(\mathcal{P}(\mathcal{F}))\log^{\frac{1}{2}}(8e^2n^3c^3)\cdot\log\frac{M\sqrt{n}}{\rho B} \quad \text{(Use inequality (8) )}$$

$$= 12\sqrt{2}\rho\sqrt{c}\widetilde{\Re}_{nc}(\mathcal{P}(\mathcal{F}))\left(1 + \log^{\frac{1}{2}}(8e^2n^3c^3)\cdot\log\frac{M\sqrt{n}}{\rho B}\right).$$

### A.3.3 Proof of Theorem 1

We upper bound the worst-case Rademacher complexity $\widetilde{\Re}_{nc}(\mathcal{P}(\mathcal{F}))$ as the following:

$$\widetilde{\Re}_{nc}(\mathcal{P}(\mathcal{F})) = \sup_{[c]\times D\in[c]\times\mathcal{X}^n}\hat{\Re}_{[c]\times D}(\mathcal{P}(\mathcal{F}))$$

$$= \sup_{[c]\times D\in[c]\times\mathcal{X}^n}\mathbb{E}_{\boldsymbol{\epsilon}}\left[\sup_{p_j(\boldsymbol{f}(\boldsymbol{x}_i))\in\mathcal{P}(\mathcal{F})}\frac{1}{nc}\sum_{i=1}^n\sum_{j=1}^c\epsilon_{ij}p_j(\boldsymbol{f}(\boldsymbol{x}_i))\right]$$

$$= \sup_{[c]\times D\in[c]\times\mathcal{X}^n}\mathbb{E}_{\boldsymbol{\epsilon}}\left[\sup_{f_j\in\mathcal{F}_j}\frac{1}{nc}\sum_{i=1}^n\sum_{j=1}^c\epsilon_{ij}f_j(\boldsymbol{x}_i)\right]$$

$$= \sup_{\|\zeta_j(\phi_j(\boldsymbol{x}_i))\|_2\leq A:i\in[n],j\in[c]}\frac{1}{nc}\mathbb{E}_{\boldsymbol{\epsilon}}\left[\sup_{\|\boldsymbol{w}_j\|_2\leq\Lambda}\sum_{i=1}^n\sum_{j=1}^c\epsilon_{ij}\langle\boldsymbol{w}_j,\zeta_j(\phi_j(\boldsymbol{x}_i))\rangle\right]$$

$$= \sup_{\|\zeta_j(\phi_j(\boldsymbol{x}_i))\|_2\leq A:i\in[n],j\in[c]}\frac{\Lambda}{nc}\mathbb{E}_{\boldsymbol{\epsilon}}\|\sum_{i=1}^n\sum_{j=1}^c\epsilon_{ij}\zeta_j(\phi_j(\boldsymbol{x}_i))\|$$

$$\leq \sup_{\|\zeta_j(\phi_j(\boldsymbol{x}_i))\|_2\leq A:i\in[n],j\in[c]}\frac{\Lambda}{nc}\left[\mathbb{E}_{\boldsymbol{\epsilon}}\|\sum_{i=1}^n\sum_{j=1}^c\epsilon_{ij}\zeta_j(\phi_j(\boldsymbol{x}_i))\|^2\right]^{\frac{1}{2}} \quad \text{(Use Jensen's Inequality)}$$

$$\leq \sup_{\|\zeta_j(\phi_j(\boldsymbol{x}_i))\|_2\leq A:i\in[n],j\in[c]}\frac{\Lambda}{nc}\left[\sum_{i=1}^n\sum_{j=1}^c\|\zeta_j(\phi_j(\boldsymbol{x}_i))\|^2\right]^{\frac{1}{2}} \leq \frac{\Lambda A}{\sqrt{nc}}. \quad \text{(Use Lemma 2)}$$

$$(9)$$

Then, we have

$$\hat{\Re}_D(\mathcal{F}) \leq 12\sqrt{2}\rho\sqrt{c}\widetilde{\Re}_{nc}(\mathcal{P}(\mathcal{F}))\left(1 + \log^{\frac{1}{2}}(8e^2n^3c^3) \cdot \log \frac{M\sqrt{n}}{\rho B}\right)$$

$$\leq \frac{12\sqrt{2}\rho\Lambda A\left(1 + \log^{\frac{1}{2}}(8e^2n^3c^3) \cdot \log \frac{M\sqrt{n}}{\rho B}\right)}{\sqrt{n}}.$$

Combining with (5), then

$$R(\boldsymbol{f}) \leq \widehat{R}_D(\boldsymbol{f}) + \frac{24\sqrt{2}\rho\Lambda A\left(1 + \log^{\frac{1}{2}}(8e^2n^3c^3) \cdot \log \frac{M\sqrt{n}}{\rho B}\right)}{\sqrt{n}} + 3M\sqrt{\frac{\log\frac{2}{\delta}}{2n}}.$$

### A.4 Generalization Bounds for Typical LSRL Methods

#### A.4.1 Proof of Theorem 2

The generalization analysis of LIFT involves the generalization analysis for $k$-means clustering. According to the definitions of $k$-means clustering in Subection 5.1, we have the empirical Rademacher complexity of a loss function space associated with the vector-valued function class $\mathcal{G}$:

$$\hat{\Re}_D(\mathcal{L}_{clu} \circ \mathcal{G}) = \mathbb{E}_{\boldsymbol{\epsilon}}\left[\sup_{\boldsymbol{g} \in \mathcal{G}} \frac{1}{n(n-1)} \sum_{i,j=1, i \neq j}^{n} \epsilon_i \ell_{clu}(\boldsymbol{g}(\boldsymbol{x}_i, \boldsymbol{x}_j))\right].$$

Rademacher complexity has proved to be a powerful data-dependent measure of hypothesis space complexity. However, since the $k$-means clustering framework involves pairwise functions, a sequence of pairs of i.i.d. individual observation in $k$-means clustering is no longer independent, which makes standard techniques in the i.i.d case for traditional Rademacher complexity inapplicable for $k$-means clustering. We convert the non-sum-of-i.i.d pairwise function to a sum-of-i.i.d form by using permutations in U-process [Clémençon et al., 2008].

We first proof the following lemma:

**Lemma 6.** *Let $q_\tau : \mathcal{X} \times \mathcal{X} \mapsto \mathbb{R}$ be real-valued functions indexed by $\tau \in T$ where $T$ is some set. If $\boldsymbol{x}_1, \ldots, \boldsymbol{x}_s$ and $\boldsymbol{x}_1', \ldots, \boldsymbol{x}_t'$ are i.i.d., $r = \min\{s, t\}$, then for any convex non-decreasing function $\psi$,*

$$\mathbb{E}\psi\left(\sup_{\tau \in T} \frac{1}{st} \sum_{i=1}^{s} \sum_{j=1}^{t} q_\tau\left(\boldsymbol{x}_i, \boldsymbol{x}_j'\right)\right) \leq \mathbb{E}\psi\left(\sup_{\tau \in T} \frac{1}{r} \sum_{i=1}^{r} q_\tau\left(\boldsymbol{x}_i, \boldsymbol{x}_i'\right)\right).$$

*Proof.* The proof of this lemma is inspired by [Clémençon et al., 2008].

$$\mathbb{E}\psi\left(\sup_{\tau \in T} \frac{1}{st} \sum_{i=1}^{s} \sum_{j=1}^{t} q_\tau\left(\boldsymbol{x}_i, \boldsymbol{x}_j'\right)\right)$$

$$= \mathbb{E}\psi\left(\sup_{\tau \in T} \frac{1}{s!} \sum_{\pi_{\boldsymbol{x}}} \frac{1}{t!} \sum_{\pi_{\boldsymbol{x}'}} \frac{1}{r} \sum_{i=1}^{r} q_\tau\left(\boldsymbol{x}_{\pi(i)}, \boldsymbol{x}_{\pi(i)}'\right)\right)$$

$$\leq \mathbb{E}\psi\left(\frac{1}{s!} \sum_{\pi_{\boldsymbol{x}}} \frac{1}{t!} \sum_{\pi_{\boldsymbol{x}'}} \sup_{\tau \in T} \frac{1}{r} \sum_{i=1}^{r} q_\tau\left(\boldsymbol{x}_{\pi(i)}, \boldsymbol{x}_{\pi(i)}'\right)\right) \quad (\psi \text{ is nondecreasing})$$

$$\leq \frac{1}{s!} \sum_{\pi_{\boldsymbol{x}}} \frac{1}{t!} \sum_{\pi_{\boldsymbol{x}'}} \mathbb{E}\psi\left(\sup_{\tau \in T} \frac{1}{r} \sum_{i=1}^{r} q_\tau\left(\boldsymbol{x}_{\pi(i)}, \boldsymbol{x}_{\pi(i)}'\right)\right) \quad (\text{Jensen's inequality})$$

$$= \mathbb{E}\psi\left(\sup_{\tau \in T} \frac{1}{r} \sum_{i=1}^{r} q_\tau\left(\boldsymbol{x}_i, \boldsymbol{x}_i'\right)\right).$$

$\square$

According to Lemma 6, $\hat{\Re}_D(\mathcal{L}_{clu} \circ \mathcal{G})$ can be bounded by

$$\hat{\Re}_{D'}(\mathcal{L}_{clu} \circ \mathcal{G}) := \mathbb{E}_{\epsilon} \left[ \sup_{\boldsymbol{g} \in \mathcal{G}} \frac{1}{\lfloor \frac{n}{2} \rfloor} \sum_{i=1}^{\lfloor \frac{n}{2} \rfloor} \epsilon_i \ell_{clu}(\boldsymbol{g}(\boldsymbol{x}_i, \boldsymbol{x}_{i+\lfloor \frac{n}{2} \rfloor})) \right],$$

where each $\epsilon_i$ is an independent Rademacher random variable.

In order to obtain tighter generalization bounds for $k$-means clustering, we develop the following novel vector-contraction inequality:

**Lemma 7.** *Let $\mathcal{G}$ be a vector-valued function class of $k$-means clustering defined in Subsection 5.1. Given a dataset $D$ of size $n$. Then, we have*

$$\hat{\Re}_D(\mathcal{L}_{clu} \circ \mathcal{G}) \leq 12\sqrt{K} \max_k \widetilde{\Re}_{\lfloor \frac{n}{2} \rfloor}(\mathcal{G}_k) \left( 1 + \log^{\frac{1}{2}}(e^2 n^3) \cdot \log \frac{M}{\widetilde{\Re}_{\lfloor \frac{n}{2} \rfloor}(\mathcal{G}_k)} \right),$$

*where $\widetilde{\Re}_{\lfloor \frac{n}{2} \rfloor}(\mathcal{G}_k)$ is the worst-case Rademacher complexity, $\mathcal{G}_k$ is the restriction of the function class along the $k$-th coordinate, $g_k \in \mathcal{G}_k$.*

*Proof.* We first introduce the following lemma:

**Lemma 8** (Lemma 1 in [Foster and Rakhlin, 2019]). *Let $\mathcal{F} \subseteq \left\{ \boldsymbol{f} : \mathcal{X} \to \mathbb{R}^K \right\}$, and let $\phi_1, \ldots, \phi_n$ each be $L$ Lipschitz with respect to the $\ell_\infty$ norm. For any $D$ with a finite sample of size $n$ and $\epsilon > 0$, we have that*

$$\log \mathcal{N}_2 (\epsilon, \phi \circ \mathcal{F}, D) \leq K \max_k \log \mathcal{N}_\infty \left( \frac{\epsilon}{L}, \mathcal{F}_k, D \right), \tag{10}$$

*where $\mathcal{F}_k$ is the restriction of the function class along the $k$-th coordinate, $f_k \in \mathcal{F}_k$, $k \in [K]$.*

The empirical $\ell_\infty$ norm covering number of $\mathcal{G}_k$ can be bounded by the fat-shattering dimension, and the fat-shattering dimension can be bounded by the worst-case Rademacher complexity of $\mathcal{G}_k$. Combined with the above steps, we have

$$\hat{\Re}_D(\mathcal{L}_{clu} \circ \mathcal{G}) \leq \hat{\Re}_{D'}(\mathcal{L}_{clu} \circ \mathcal{G})$$

$$\leq \inf_{\alpha > 0} \left( 4\alpha + \frac{12}{\sqrt{n}} \int_\alpha^M \sqrt{\log \mathcal{N}_2(\epsilon, \mathcal{L}_{clu} \circ \mathcal{G}, D')} d\epsilon \right) \quad \text{(Use Lemma 5 )}$$

$$\leq \inf_{\alpha > 0} \left( 4\alpha + \frac{12}{\sqrt{n}} \int_\alpha^M \sqrt{K \max_k \log \mathcal{N}_\infty(\epsilon, \mathcal{G}_k, D')} d\epsilon \right)$$

(Use inequality (10) and $\ell_{clu}(\cdot)$ is 1-Lipschitz w.r.t. $\ell_\infty$ norm for $k$-means clustering)

$$\leq \inf_{\alpha > 0} \left( 4\alpha + \frac{12}{\sqrt{n}} \int_\alpha^M \sqrt{K \max_k \text{fat}_\epsilon(\mathcal{G}_k) \log \frac{2eB\lfloor \frac{n}{2} \rfloor}{\epsilon}} d\epsilon \right) \quad \text{(Use Lemma 4)}$$

$$\leq \inf_{\alpha > 0} \left( 4\alpha + \frac{12}{\sqrt{n}} \int_\alpha^M \sqrt{K \max_k \frac{4\lfloor \frac{n}{2} \rfloor \hat{\Re}_{D'}^2(\mathcal{G}_k)}{\epsilon^2} \log \frac{2eG\lfloor \frac{n}{2} \rfloor}{\hat{\Re}_{D'}(\mathcal{G}_k)}} d\epsilon \right) \quad \text{(Use Lemma 3)}$$

$$\leq \inf_{\alpha > 0} \left( 4\alpha + \frac{12}{\sqrt{n}} \int_\alpha^M \sqrt{K \max_k \frac{4\lfloor \frac{n}{2} \rfloor \widetilde{\Re}_{\lfloor \frac{n}{2} \rfloor}^2(\mathcal{G}_k)}{\epsilon^2} \log \frac{2eG\lfloor \frac{n}{2} \rfloor}{\widetilde{\Re}_{\lfloor \frac{n}{2} \rfloor}(\mathcal{G}_k)}} d\epsilon \right)$$

(The definition of the worst-case Rademacher complexity)

$$\leq \inf_{\alpha > 0} \left( 4\alpha + \frac{12}{\sqrt{n}} \int_\alpha^M \sqrt{\frac{2nK \max_k \widetilde{\Re}_{\lfloor \frac{n}{2} \rfloor}^2(\mathcal{G}_k)}{\epsilon^2} \log(en^{\frac{3}{2}})} d\epsilon \right)$$

(Use the similar technique to the proof in inequality (8), the lower bound of $\widetilde{\Re}_{\lfloor\frac{n}{2}\rfloor}(\mathcal{G}_k) \geq \dfrac{G}{\sqrt{2\lfloor\frac{n}{2}\rfloor}}$ )

$$\leq \inf_{\alpha>0}\left(4\alpha + 12\sqrt{K}\max_k \widetilde{\Re}_{\lfloor\frac{n}{2}\rfloor}(\mathcal{G}_k)\log^{\frac{1}{2}}(e^2n^3)\int_\alpha^M \epsilon^{-1}d\epsilon\right)$$

$$\leq 12\sqrt{K}\max_k \widetilde{\Re}_{\lfloor\frac{n}{2}\rfloor}(\mathcal{G}_k) + 12\sqrt{K}\max_k \widetilde{\Re}_{\lfloor\frac{n}{2}\rfloor}(\mathcal{G}_k)\log^{\frac{1}{2}}(e^2n^3)\cdot\log\frac{M}{3\sqrt{K}\max_k \widetilde{\Re}_{\lfloor\frac{n}{2}\rfloor}(\mathcal{G}_k)}$$

(Choose $\alpha = 3\sqrt{K}\max_k \widetilde{\Re}_{\lfloor\frac{n}{2}\rfloor}(\mathcal{G}_k)$)

$$= 12\sqrt{K}\max_k \widetilde{\Re}_{\lfloor\frac{n}{2}\rfloor}(\mathcal{G}_k)\left(1 + \log^{\frac{1}{2}}(e^2n^3)\cdot\log\frac{M}{\widetilde{\Re}_{\lfloor\frac{n}{2}\rfloor}(\mathcal{G}_k)}\right).$$

$\square$

Since $k$-means clustering-based LSRL method is two-stage, the $k$-means clustering is used to generate label-specific representations in the first stage, and the second stage is conventional multi-label learning. Therefore, the corresponding whole function class is actually denoted as $\mathcal{H} = \mathcal{F} + \mathcal{L}_{clu}\circ\mathcal{G}$. Since $\widetilde{\Re}_{nc}(\mathcal{P}(\mathcal{H})) \leq \widetilde{\Re}_{nc}(\mathcal{P}(\mathcal{F})) + \widetilde{\Re}_{nc}(\mathcal{P}(\mathcal{L}_{clu}\circ\mathcal{G})$ [Bartlett and Mendelson, 2002], with Lemma 1, we have

$$\hat{\Re}_D(\mathcal{L}\circ\mathcal{H}) \leq 12\sqrt{2}\rho\sqrt{c}\left(\widetilde{\Re}_{nc}(\mathcal{P}(\mathcal{F})) + \widetilde{\Re}_{nc}(\mathcal{P}(\mathcal{L}_{clu}\circ\mathcal{G}))\right)\left(1 + \log^{\frac{1}{2}}(8e^2n^3c^3)\cdot\log\frac{M\sqrt{n}}{\mu B}\right).$$

According to Lemma 7, we have that the worst-case Rademacher complexity of the loss function space associated with $\mathcal{P}(\mathcal{G})$ can be bounded by the worst-case Rademacher complexity of the restriction of the function class along each coordinate $\mathcal{P}(\mathcal{G}_k)$. Hence, for $k$-means clustering-based LSRL method, Lemma 7 involves the upper and lower bounds of $\widetilde{\Re}_{\lfloor\frac{nc}{2}\rfloor}(\mathcal{P}(\mathcal{G}_k))$.

We then obtain the lower bound of the worst-case Rademacher complexity $\widetilde{\Re}_{\lfloor\frac{nc}{2}\rfloor}(\mathcal{P}(\mathcal{G}_k))$ by the Khintchine-Kahane inequality with $p=1$:

$$\widetilde{\Re}_{\lfloor\frac{nc}{2}\rfloor}(\mathcal{P}(\mathcal{G}_k)) = \sup_{[c]\times D'\in[c]\times\mathcal{X}^{\lfloor\frac{n}{2}\rfloor}}\hat{\Re}_{[c]\times D'}(\mathcal{P}(\mathcal{G}_k))$$

$$= \sup_{[c]\times D'\in[c]\times\mathcal{X}^{\lfloor\frac{n}{2}\rfloor}}\mathbb{E}_{\boldsymbol{\epsilon}}\left[\sup_{p_j(g_k(\boldsymbol{x}_i,\boldsymbol{x}_{i+\lfloor\frac{n}{2}\rfloor}))\in\mathcal{P}(\mathcal{G}_k)}\frac{1}{\lfloor\frac{n}{2}\rfloor}\sum_{i=1}^{\lfloor\frac{n}{2}\rfloor}\frac{1}{c}\sum_{j=1}^c\epsilon_{ij}p_j(g_k(\boldsymbol{x}_i,\boldsymbol{x}_{i+\lfloor\frac{n}{2}\rfloor}))\right]$$

$$= \sup_{[c]\times D'\in[c]\times\mathcal{X}^{\lfloor\frac{n}{2}\rfloor}}\mathbb{E}_{\boldsymbol{\epsilon}}\left[\sup_{g_k^j\in\mathcal{G}_k^j}\frac{1}{\lfloor\frac{n}{2}\rfloor c}\sum_{i=1}^{\lfloor\frac{n}{2}\rfloor}\sum_{j=1}^c\epsilon_{ij}g_k^j(\boldsymbol{x}_i,\boldsymbol{x}_{i+\lfloor\frac{n}{2}\rfloor})\right]$$

$$= \sup_{\|Z_k^j(\boldsymbol{x}_i,\boldsymbol{x}_{i+\lfloor\frac{n}{2}\rfloor})\|\leq1}\mathbb{E}_{\boldsymbol{\epsilon}}\left[\sup_{\|V(\boldsymbol{x}_i,\boldsymbol{x}_{i+\lfloor\frac{n}{2}\rfloor})\|\leq4R^2}\frac{1}{\lfloor\frac{n}{2}\rfloor c}\sum_{i=1}^{\lfloor\frac{n}{2}\rfloor}\sum_{j=1}^c\epsilon_{ij}V(\boldsymbol{x}_i,\boldsymbol{x}_{i+\lfloor\frac{n}{2}\rfloor})Z_k^j(\boldsymbol{x}_i,\boldsymbol{x}_{i+\lfloor\frac{n}{2}\rfloor})\right]$$

$$= \sup_{\|Z_k^j(\boldsymbol{x}_i,\boldsymbol{x}_{i+\lfloor\frac{n}{2}\rfloor})\|\leq1}\frac{4R^2}{\lfloor\frac{n}{2}\rfloor c}\mathbb{E}_{\boldsymbol{\epsilon}}\|\sum_{i=1}^{\lfloor\frac{n}{2}\rfloor}\sum_{j=1}^c\epsilon_{ij}Z_k^j(\boldsymbol{x}_i,\boldsymbol{x}_{i+\lfloor\frac{n}{2}\rfloor})\|$$

$$\geq \sup_{\|Z_k^j(\boldsymbol{x}_i,\boldsymbol{x}_{i+\lfloor\frac{n}{2}\rfloor})\|\leq1}\frac{4R^2}{\lfloor\frac{n}{2}\rfloor c}\frac{1}{\sqrt{2}}\left[\sum_{i=1}^{\lfloor\frac{n}{2}\rfloor}\sum_{j=1}^c\|Z_k^j(\boldsymbol{x}_i,\boldsymbol{x}_{i+\lfloor\frac{n}{2}\rfloor})\|^2\right]^{\frac{1}{2}}.$$

Since $\|Z_k^j(\boldsymbol{x}_i,\boldsymbol{x}_{i+\lfloor\frac{n}{2}\rfloor})\| \leq 1$, we set $\sup_{\|Z_k^j(\boldsymbol{x}_i,\boldsymbol{x}_{i+\lfloor\frac{n}{2}\rfloor})\|\leq1}\frac{1}{\lfloor\frac{n}{2}\rfloor c}\left[\sum_{i=1}^{\lfloor\frac{n}{2}\rfloor}\sum_{j=1}^c\|Z_k^j(\boldsymbol{x}_i,\boldsymbol{x}_{i+\lfloor\frac{n}{2}\rfloor})\|^2\right]^{\frac{1}{2}} = \frac{1}{\sqrt{\lfloor\frac{n}{2}\rfloor c}}$. So, $\widetilde{\Re}_{\lfloor\frac{nc}{2}\rfloor}(\mathcal{P}(\mathcal{G}_k))) \geq \frac{4R^2}{\sqrt{nc}} = \frac{G}{\sqrt{nc}}$.

Then, we replace the lower bound of $\widetilde{\Re}_{\lfloor \frac{n}{2} \rfloor}(\mathcal{G}_k)$ in the proof of Lemma 7 with the lower bound of $\widetilde{\Re}_{\lfloor \frac{nc}{2} \rfloor}(\mathcal{P}(\mathcal{G}_k)))$, and we have

$$\widetilde{\Re}_{nc}(\mathcal{P}(\mathcal{L}_{clu} \circ \mathcal{G})) \leq 12\sqrt{K} \max_k \widetilde{\Re}_{\lfloor \frac{nc}{2} \rfloor}(\mathcal{P}(\mathcal{G}_k))) \left( 1 + \log^{\frac{1}{2}}(e^2 n^3 c^3) \cdot \log \frac{M\sqrt{nc}}{G} \right).$$

We then upper bound the worst-case Rademacher complexity $\widetilde{\Re}_{\lfloor \frac{nc}{2} \rfloor}(\mathcal{P}(\mathcal{G}_k))$ as the following:

$$\widetilde{\Re}_{\lfloor \frac{nc}{2} \rfloor}(\mathcal{P}(\mathcal{G}_k))$$

$$= \sup_{[c] \times D' \in [c] \times \mathcal{X}^{\lfloor \frac{n}{2} \rfloor}} \hat{\Re}_{[c] \times D'}(\mathcal{P}(\mathcal{G}_k))$$

$$= \sup_{[c] \times D' \in [c] \times \mathcal{X}^{\lfloor \frac{n}{2} \rfloor}} \mathbb{E}_{\boldsymbol{\epsilon}} \left[ \sup_{p_j(g_k(\boldsymbol{x}_i, \boldsymbol{x}_{i+\lfloor \frac{n}{2} \rfloor})) \in \mathcal{P}(\mathcal{G}_k)} \frac{1}{\lfloor \frac{n}{2} \rfloor} \sum_{i=1}^{\lfloor \frac{n}{2} \rfloor} \frac{1}{c} \sum_{j=1}^{c} \epsilon_{ij} p_j(g_k(\boldsymbol{x}_i, \boldsymbol{x}_{i+\lfloor \frac{n}{2} \rfloor})) \right]$$

$$= \sup_{[c] \times D' \in [c] \times \mathcal{X}^{\lfloor \frac{n}{2} \rfloor}} \mathbb{E}_{\boldsymbol{\epsilon}} \left[ \sup_{g_k^j \in \mathcal{G}_k^j} \frac{1}{\lfloor \frac{n}{2} \rfloor c} \sum_{i=1}^{\lfloor \frac{n}{2} \rfloor} \sum_{j=1}^{c} \epsilon_{ij} g_k^j(\boldsymbol{x}_i, \boldsymbol{x}_{i+\lfloor \frac{n}{2} \rfloor}) \right]$$

$$= \sup_{\|Z_k^j(\boldsymbol{x}_i, \boldsymbol{x}_{i+\lfloor \frac{n}{2} \rfloor})\| \leq 1} \mathbb{E}_{\boldsymbol{\epsilon}} \left[ \sup_{\|V(\boldsymbol{x}_i, \boldsymbol{x}_{i+\lfloor \frac{n}{2} \rfloor})\| \leq 4R^2} \frac{1}{\lfloor \frac{n}{2} \rfloor c} \sum_{i=1}^{\lfloor \frac{n}{2} \rfloor} \sum_{j=1}^{c} \epsilon_{ij} V(\boldsymbol{x}_i, \boldsymbol{x}_{i+\lfloor \frac{n}{2} \rfloor}) Z_k^j(\boldsymbol{x}_i, \boldsymbol{x}_{i+\lfloor \frac{n}{2} \rfloor}) \right]$$

$$= \sup_{\|Z_k^j(\boldsymbol{x}_i, \boldsymbol{x}_{i+\lfloor \frac{n}{2} \rfloor})\| \leq 1} \frac{4R^2}{\lfloor \frac{n}{2} \rfloor c} \mathbb{E}_{\boldsymbol{\epsilon}} \| \sum_{i=1}^{\lfloor \frac{n}{2} \rfloor} \sum_{j=1}^{c} \epsilon_{ij} Z_k^j(\boldsymbol{x}_i, \boldsymbol{x}_{i+\lfloor \frac{n}{2} \rfloor}) \|$$

$$\leq \sup_{\|Z_k^j(\boldsymbol{x}_i, \boldsymbol{x}_{i+\lfloor \frac{n}{2} \rfloor})\| \leq 1} \frac{4R^2}{\lfloor \frac{n}{2} \rfloor c} \left[ \mathbb{E}_{\boldsymbol{\epsilon}} \| \sum_{i=1}^{\lfloor \frac{n}{2} \rfloor} \sum_{j=1}^{c} \epsilon_{ij} Z_k^j(\boldsymbol{x}_i, \boldsymbol{x}_{i+\lfloor \frac{n}{2} \rfloor}) \|^2 \right]^{\frac{1}{2}}$$

$$\leq \sup_{\|Z_k^j(\boldsymbol{x}_i, \boldsymbol{x}_{i+\lfloor \frac{n}{2} \rfloor})\| \leq 1} \frac{4R^2}{\lfloor \frac{n}{2} \rfloor c} \left[ \sum_{i=1}^{\lfloor \frac{n}{2} \rfloor} \sum_{j=1}^{c} \| Z_k^j(\boldsymbol{x}_i, \boldsymbol{x}_{i+\lfloor \frac{n}{2} \rfloor}) \|^2 \right]^{\frac{1}{2}}$$

$$\leq \frac{G}{\sqrt{\lfloor \frac{n}{2} \rfloor c}}$$

$$\leq \frac{2G}{\sqrt{nc}}.$$

Hence, we have

$$\widetilde{\Re}_{nc}(\mathcal{P}(\mathcal{L}_{clu} \circ \mathcal{G})) \leq \frac{24\sqrt{K}G}{\sqrt{nc}} \left( 1 + \log^{\frac{1}{2}}(e^2 n^3 c^3) \cdot \log \frac{M\sqrt{nc}}{G} \right).$$

Use the similar technique to the proof of the inequality above, the upper bound of $\widetilde{\Re}_{nc}(\mathcal{P}(\mathcal{F}))$ is $\frac{2\kappa \Lambda \sqrt{K}R}{\sqrt{nc}}$, since $\|\phi_j(\boldsymbol{x})\| = \sqrt{\sum_{k=1}^{K}(d(\boldsymbol{x}, \boldsymbol{c}_k^j))^2} \leq \sqrt{K} \max_k \|\boldsymbol{x} - \boldsymbol{c}_k^j\| \leq \sqrt{K} \max_k (\|\boldsymbol{x}\| + \|\boldsymbol{c}_k^j\|) \leq 2\sqrt{K}R$.

Finally, combining with the above inequalities and (5), we have

$$R(\boldsymbol{f}) \leq \widehat{R}_D(\boldsymbol{f}) + 3M\sqrt{\frac{\log \frac{2}{\delta}}{2n}} + \frac{48\sqrt{2}\rho\kappa\Lambda\sqrt{K}R \left( 1 + \log^{\frac{1}{2}}(8e^2 n^3 c^3) \cdot \log \frac{M\sqrt{n}}{\mu B} \right)}{\sqrt{n}}$$

$$+ \frac{24^2\sqrt{2}\rho\sqrt{K}G}{\sqrt{n}} \left( 1 + \log^{\frac{1}{2}}(e^2 n^3 c^3) \cdot \log \frac{M\sqrt{nc}}{G} \right) \left( 1 + \log^{\frac{1}{2}}(8e^2 n^3 c^3) \cdot \log \frac{M\sqrt{n}}{\mu B} \right).$$

### A.4.2   Proof of Proposition 2

First, we prove that the squared loss is 1 pseudo-Lipschitz of order 2.

$$
\begin{aligned}
&\left| \ell_b\left(y_j f_j(\boldsymbol{x})\right) - \ell_b\left(y_j f'_j(\boldsymbol{x})\right) \right| \\
&= \left| \left(y_j - f_j(\boldsymbol{x})\right)^2 - \left(y_j - f'_j(\boldsymbol{x})\right)^2 \right| \\
&= \left| \left(y_j - f_j(\boldsymbol{x})\right) + \left(y_j - f'_j(\boldsymbol{x})\right) \right| \cdot \left| \left(y_j - f_j(\boldsymbol{x})\right) - \left(y_j - f'_j(\boldsymbol{x})\right) \right| \\
&\leq \left( |y_j - f_j(\boldsymbol{x})| + |y_j - f'_j(\boldsymbol{x})| \right) \cdot \left| \left(y_j - f_j(\boldsymbol{x})\right) - \left(y_j - f'_j(\boldsymbol{x})\right) \right| \\
&\leq \left( 1 + |y_j - f_j(\boldsymbol{x})| + |y_j - f'_j(\boldsymbol{x})| \right) \cdot \left| \left(y_j - f_j(\boldsymbol{x})\right) - \left(y_j - f'_j(\boldsymbol{x})\right) \right|
\end{aligned}
$$

According to the definition of the pseudo-Lipschitz function, the squared loss is 1 pseudo-Lipschitz of order 2. Then, we have

$$
\begin{aligned}
&\left| \ell_b\left(y_j f_j(\boldsymbol{x})\right) - \ell_b\left(y_j f'_j(\boldsymbol{x})\right) \right| \\
&\leq \left( 1 + |y_j - f_j(\boldsymbol{x})| + |y_j - f'_j(\boldsymbol{x})| \right) \cdot \left| \left(y_j - f_j(\boldsymbol{x})\right) - \left(y_j - f'_j(\boldsymbol{x})\right) \right| \\
&\leq \left( 1 + 2|y_j| + |f_j(\boldsymbol{x})| + |f'_j(\boldsymbol{x})| \right) \left| f_j(\boldsymbol{x}) - f'_j(\boldsymbol{x}) \right| \\
&\leq (3 + 2B) \left| f_j(\boldsymbol{x}) - f'_j(\boldsymbol{x}) \right|.
\end{aligned}
$$

Second, we prove that the surrogate Decomposable Loss is $(3 + 2B)c$-Lipschitz continuous with respect to the $\ell_\infty$ norm if the base loss $\ell_b$ is the squared loss.

$$
\begin{aligned}
&\left| \ell_D(\boldsymbol{f}(\boldsymbol{x}), \boldsymbol{y}) - \ell_D\left(\boldsymbol{f}'(\boldsymbol{x}), \boldsymbol{y}\right) \right| \\
&= \left| \sum_{j=1}^{c} \ell_b\left(y_j f_j(\boldsymbol{x})\right) - \sum_{j=1}^{c} \ell_b\left(y_j f'_j(\boldsymbol{x})\right) \right| \\
&= \sum_{j=1}^{c} \left| \ell_b\left(y_j f_j(\boldsymbol{x})\right) - \ell_b\left(y_j f'_j(\boldsymbol{x})\right) \right| \\
&\leq \sum_{j=1}^{c} (3 + 2B) \left| f_j(\boldsymbol{x}) - f'_j(\boldsymbol{x}) \right| \\
&\leq c(3 + 2B) \max_{j \in [c]} \left| f_j(\boldsymbol{x}) - f'_j(\boldsymbol{x}) \right| \\
&= (3 + 2B)c \left\| \boldsymbol{f}(\boldsymbol{x}) - \boldsymbol{f}'(\boldsymbol{x}) \right\|_\infty.
\end{aligned}
$$

### A.4.3   Proof of Theorem 3

Compared with the function class of LSRL where each label corresponds to a Lasso, the function class of LLSF introduces the additional sharing constraint. In fact, the introduction of the sharing constraint reduces the complexity of the function class compared with the original class. Then, the complexity for the function class of LLSF can be bounded by the complexity of the function class of LSRL where each label corresponds to a Lasso. Hence, the complexity analysis of the function class of LLSF can be converted into giving the bound of the Rademacher complexity of the LSRL function class where each label corresponds to a Lasso.

According to the definition, the function class of LSRL where each label corresponds to a Lasso means that in the class of LSRL defined by (1), the base loss $\ell_b$ is the squared loss, the nonlinear mappings $\zeta(\cdot)$ and $\phi(\cdot)$ are both identity transformations for any $j \in [c]$, and the constraint $\alpha(\boldsymbol{w})$ is $\|\boldsymbol{w}_j\|_1 \leq \Lambda$ for any $j \in [c]$. The proof process is similar to Lemma 1 and Theorem 1, but the upper bound of the worst-case Rademacher complexity $\widetilde{\Re}_{nc}(\mathcal{P}(\mathcal{F}))$ here is different from Theorem 1. According to the above definitions, we have

$$\widetilde{\Re}_{nc}(\mathcal{P}(\mathcal{F}))$$

$$= \sup_{[c]\times D\in[c]\times\mathcal{X}^n} \hat{\Re}_{[c]\times D}(\mathcal{P}(\mathcal{F}))$$

$$= \sup_{[c]\times D\in[c]\times\mathcal{X}^n} \mathbb{E}_{\boldsymbol{\epsilon}}\left[\sup_{p_j(\boldsymbol{f}(\boldsymbol{x}_i))\in\mathcal{P}(\mathcal{F})} \frac{1}{nc}\sum_{i=1}^{n}\sum_{j=1}^{c}\epsilon_{ij}p_j(\boldsymbol{f}(\boldsymbol{x}_i))\right]$$

$$= \sup_{[c]\times D\in[c]\times\mathcal{X}^n} \mathbb{E}_{\boldsymbol{\epsilon}}\left[\sup_{f_j\in\mathcal{F}_j} \frac{1}{nc}\sum_{i=1}^{n}\sum_{j=1}^{c}\epsilon_{ij}f_j(\boldsymbol{x}_i)\right]$$

$$= \sup_{\|\boldsymbol{x}_i^j\|_2\le R: i\in[n],j\in[c]} \frac{1}{nc}\mathbb{E}_{\boldsymbol{\epsilon}}\left[\sup_{\|\boldsymbol{w}_j\|_1\le\Lambda} \sum_{i=1}^{n}\sum_{j=1}^{c}\epsilon_{ij}\langle\boldsymbol{w}_j,\boldsymbol{x}_i^j\rangle\right]$$

$$= \sup_{\|\boldsymbol{x}_i^j\|_2\le R: i\in[n],j\in[c]} \frac{1}{nc}\mathbb{E}_{\boldsymbol{\epsilon}}\left[\sup_{\|\boldsymbol{w}_j\|_1\le\Lambda} \|\boldsymbol{w}_j\|_1\|\sum_{i=1}^{n}\sum_{j=1}^{c}\epsilon_{ij}\boldsymbol{x}_i^j\|_\infty\right] \quad \text{(Use Hölder's Inequality)}$$

$$\le \sup_{\|\boldsymbol{x}_i^j\|_2\le R: i\in[n],j\in[c]} \frac{\Lambda}{nc}\mathbb{E}_{\boldsymbol{\epsilon}}\|\sum_{i=1}^{n}\sum_{j=1}^{c}\epsilon_{ij}\boldsymbol{x}_i^j\|_2$$

$$\le \sup_{\|\boldsymbol{x}_i^j\|_2\le R: i\in[n],j\in[c]} \frac{\Lambda}{nc}\left[\mathbb{E}_{\boldsymbol{\epsilon}}\|\sum_{i=1}^{n}\sum_{j=1}^{c}\epsilon_{ij}\boldsymbol{x}_i^j\|_2^2\right]^{\frac{1}{2}} \quad \text{(Use Jensen's Inequality)}$$

$$\le \sup_{\|\boldsymbol{x}_i^j\|_2\le R: i\in[n],j\in[c]} \frac{\Lambda}{nc}\left[\sum_{i=1}^{n}\sum_{j=1}^{c}\|\boldsymbol{x}_i^j\|_2^2\right]^{\frac{1}{2}} \quad \text{(Use Lemma 2)}$$

$$\le \frac{\Lambda R}{\sqrt{nc}}. \tag{11}$$

Combining with Lemma 1, inequalities (11), (5), and $\rho=(3+2B)c$, then we have

$$R(\boldsymbol{f}) \le \widehat{R}_D(\boldsymbol{f}) + \frac{24\sqrt{2}(3+2B)c\Lambda R\left(1+\log^{\frac{1}{2}}(8e^2n^3c^3)\cdot\log\frac{M\sqrt{n}}{\rho B}\right)}{\sqrt{n}} + 3M\sqrt{\frac{\log\frac{2}{\delta}}{2n}}.$$

### A.4.4 Proof of Theorem 4

First, we upper bound the worst-case Rademacher complexity $\widetilde{\Re}_{nc}(\mathcal{P}(\mathcal{F}))$ for DNN-based LSRL method. With the definitions in the main paper, we have

$$\widetilde{\Re}_{nc}(\mathcal{P}(\mathcal{F}))$$

$$= \sup_{[c]\times D\in[c]\times\mathcal{X}^n} \hat{\Re}_{[c]\times D}(\mathcal{P}(\mathcal{F}))$$

$$= \sup_{[c]\times D\in[c]\times\mathcal{X}^n} \mathbb{E}_{\boldsymbol{\epsilon}}\left[\sup_{p_j(\boldsymbol{f}(\boldsymbol{x}_i))\in\mathcal{P}(\mathcal{F})} \frac{1}{nc}\sum_{i=1}^{n}\sum_{j=1}^{c}\epsilon_{ij}p_j(\boldsymbol{f}(\boldsymbol{x}_i))\right]$$

$$= \sup_{[c]\times D\in[c]\times\mathcal{X}^n} \mathbb{E}_{\boldsymbol{\epsilon}}\left[\sup_{f_j\in\mathcal{F}_j} \frac{1}{nc}\sum_{i=1}^{n}\sum_{j=1}^{c}\epsilon_{ij}f_j(\boldsymbol{x}_i)\right]$$

$$= \sup_{[c]\times D\in[c]\times\mathcal{X}^n} \mathbb{E}_{\boldsymbol{\epsilon}}\left[\sup_{\|\boldsymbol{w}_j\|\le\Lambda} \frac{1}{nc}\sum_{i=1}^{n}\sum_{j=1}^{c}\epsilon_{ij}\sigma_{sig}(\boldsymbol{w}_j^\top\phi_j(\boldsymbol{x}_i))\right]$$

$$\leq 2 \cdot \frac{1}{4} \sup_{[c] \times D \in [c] \times \mathcal{X}^n} \mathbb{E}_{\boldsymbol{\epsilon}} \left[ \sup_{\|\boldsymbol{w}_j\| \leq \Lambda, \phi_j} \frac{1}{nc} \sum_{i=1}^{n} \sum_{j=1}^{c} \epsilon_{ij} \boldsymbol{w}_j^\top \phi_j(\boldsymbol{x}_i) \right]$$

(The Lipschitz constant of sigmoid activation is bounded by $\frac{1}{4}$)

$$\leq \frac{1}{2} \Lambda \sup_{[c] \times D \in [c] \times \mathcal{X}^n} \mathbb{E}_{\boldsymbol{\epsilon}} \sup_{\phi_j} \frac{1}{nc} \left\| \sum_{i=1}^{n} \sum_{j=1}^{c} \epsilon_{ij} \phi_j(\boldsymbol{x}_i) \right\|$$

$$\leq \frac{1}{2} \Lambda \sup_{[c] \times D \in [c] \times \mathcal{X}^n} \mathbb{E}_{\boldsymbol{\epsilon}} \sup \frac{1}{nc} \left\| \sum_{i=1}^{n} \sum_{j=1}^{c} \epsilon_{ij} \sigma_{ReLU} \left\{ W_5 \cdot [\sigma_{ReLU}(W_4 \boldsymbol{x}_i) \odot \sigma_{sig}(W_3 \psi(Y)_j)] \right\} \right\|$$

$$\leq 2 \cdot \frac{1}{2} \Lambda \sup_{[c] \times D \in [c] \times \mathcal{X}^n} \mathbb{E}_{\boldsymbol{\epsilon}} \sup_{\|W_5\| \leq D} \frac{1}{nc} \left\| \sum_{i=1}^{n} \sum_{j=1}^{c} \epsilon_{ij} W_5 \cdot [\sigma_{ReLU}(W_4 \boldsymbol{x}_i) \odot \sigma_{sig}(W_3 \psi(Y)_j)] \right\|$$

(ReLU activation is 1-Lipschitz)

$$\leq \Lambda D \sup_{[c] \times D \in [c] \times \mathcal{X}^n} \mathbb{E}_{\boldsymbol{\epsilon}} \sup \frac{1}{nc} \left\| \sum_{i=1}^{n} \sum_{j=1}^{c} \epsilon_{ij} [\sigma_{ReLU}(W_4 \boldsymbol{x}_i) \odot \sigma_{sig}(W_3 \psi(Y)_j)] \right\|$$

$$\leq \Lambda D \sup \|\sigma_{sig}(W_3 \psi(Y)_j)\| \sup_{[c] \times D \in [c] \times \mathcal{X}^n} \mathbb{E}_{\boldsymbol{\epsilon}} \sup_{\|W_4\| \leq D} \frac{1}{nc} \left\| \sum_{i=1}^{n} \sum_{j=1}^{c} \epsilon_{ij} \sigma_{ReLU}(W_4 \boldsymbol{x}_i) \right\|$$

$$\leq 2 \Lambda D \sup \|\sigma_{sig}(W_3 \psi(Y)_j)\| \sup_{\|\boldsymbol{x}_i\| \leq R} \mathbb{E}_{\boldsymbol{\epsilon}} \sup_{\|W_4\| \leq D} \frac{1}{nc} \left\| \sum_{i=1}^{n} \sum_{j=1}^{c} \epsilon_{ij} W_4 \boldsymbol{x}_i \right\|$$

$$\leq 2 \Lambda D^2 \sup \|\sigma_{sig}(W_3 \psi(Y)_j)\| \sup_{\|\boldsymbol{x}_i\| \leq R} \mathbb{E}_{\boldsymbol{\epsilon}} \frac{1}{nc} \left\| \sum_{i=1}^{n} \sum_{j=1}^{c} \epsilon_{ij} \boldsymbol{x}_i \right\|$$

$$\leq 2 \Lambda D^2 \sup \|\sigma_{sig}(W_3 \psi(Y)_j)\| \sup_{\|\boldsymbol{x}_i\| \leq R} \frac{1}{nc} \left( \mathbb{E}_{\boldsymbol{\epsilon}} \left\| \sum_{i=1}^{n} \sum_{j=1}^{c} \epsilon_{ij} \boldsymbol{x}_i \right\|^2 \right)^{\frac{1}{2}} \quad \text{(Use Jensen's Inequality)}$$

$$\leq 2 \Lambda D^2 \sup \|\sigma_{sig}(W_3 \psi(Y)_j)\| \sup_{\|\boldsymbol{x}_i\| \leq R} \frac{1}{nc} \left( \sum_{i=1}^{n} \sum_{j=1}^{c} \|\boldsymbol{x}_i\|^2 \right)^{\frac{1}{2}}$$

$$\leq 2 \Lambda D^2 \sup \|\sigma_{sig}(W_3 \psi(Y)_j)\| \frac{R}{\sqrt{nc}}. \tag{12}$$

Then, we have to bound $\sup \|\sigma_{sig}(W_3 \psi(Y)_j)\|$,

$$\sup \|\sigma_{sig}(W_3 \psi(Y)_j)\|$$

$$\leq \frac{1}{4} \sup_{\|W_3\| \leq D} \|W_3 \psi(Y)_j\| \quad \text{(The Lipschitz constant of sigmoid activation is bounded by } \frac{1}{4})$$

$$\leq \frac{1}{4} D \sup_j \|\psi(Y)_j\| \quad \text{(Use Cauchy-Schwarz Inequality)}$$

$$= \frac{D}{4} \sup_j \sup_{\|W_2\| \leq D} \|\sigma_{ReLU}(\tilde{A} \sigma_{ReLU}(\tilde{A} Y_{j*} W_1) W_2)\|$$

$$\leq \frac{D}{4} \sup_j \sup_{\|W_2\| \leq D} \|\tilde{A} \sigma_{ReLU}(\tilde{A} Y_{j*} W_1)\| \|W_2\|$$

$$\leq \frac{D^2}{4} \sup_j \|\tilde{A}\sigma_{ReLU}(\tilde{A}Y_{j*}W_1)\|$$

$$\leq \frac{D^2}{4} \sup_j \|\sum_{i=1}^{c} \tilde{A}_{ji}\sigma_{ReLU}(\sum_{j=1}^{c} \tilde{A}_{ij}Y_{j*}W_1)\|$$

$$\leq \frac{D^2}{4} \sup_j \sum_{i=1}^{c} \tilde{A}_{ji}\|\sigma_{ReLU}(\sum_{j=1}^{c} \tilde{A}_{ij}Y_{j*}W_1)\|$$

$$\leq \frac{D^2}{4} \sup_j \sup_{\|W_1\| \leq D} \sum_{i=1}^{c} \tilde{A}_{ji} \cdot \|\sum_{j=1}^{c} \tilde{A}_{ij}Y_{j*}W_1\|$$

$$\leq \frac{D^3}{4} \sup_j \sum_{i=1}^{c} \tilde{A}_{ji} \cdot \|\sum_{j=1}^{c} \tilde{A}_{ij}Y_{j*}\| \quad \text{(Use Cauchy-Schwarz Inequality)}$$

$$\leq \frac{D^3 R}{4} \sup_j \sum_{i=1}^{c} \tilde{A}_{ji} \cdot \sum_{j=1}^{c} \tilde{A}_{ij}$$

$$\leq \frac{D^3 R}{4} \|\tilde{A}\|_\infty^2 \tag{13}$$

We denote $N(j)$ as the index set of the one-hop neighbors of the $j$-th node, and denote $g$ as the node degree, $g_{\max}$ as the maximum node degree. Then, we bound $\|\tilde{A}\|_\infty$ as follows

$$\sum_{j=1}^{n} \tilde{A}_{ij} = \sum_{j=1}^{n} \frac{A_{ij}}{\sqrt{g_i + 1}\sqrt{g_j + 1}}$$

$$= \frac{1}{\sqrt{g_i + 1}} \left( \frac{1}{\sqrt{g_i + 1}} + \sum_{j \in N(i)} \frac{1}{\sqrt{g_j + 1}} \right)$$

$$\leq \frac{1}{\sqrt{g_i + 1}} \left( \frac{1}{\sqrt{1 + 1}} + \sum_{j \in N(i)} \frac{1}{\sqrt{1 + 1}} \right)$$

$$\leq \frac{1}{\sqrt{g_i + 1}} \frac{g_i + 1}{\sqrt{2}} = \sqrt{\frac{g_i + 1}{2}} \leq \sqrt{\frac{g_{\max} + 1}{2}}. \tag{14}$$

Combining with Lemma 1, inequalities (12), (13), (14), and (5), then we have

$$R(\boldsymbol{f}) \leq \widehat{R}_D(\boldsymbol{f}) + \frac{6\sqrt{2}\rho\Lambda D^5 R^2 (g_{\max} + 1)\left(1 + \log^{\frac{1}{2}}(8e^2n^3c^3) \cdot \log\frac{M\sqrt{n}}{\rho B}\right)}{\sqrt{n}} + 3M\sqrt{\frac{\log\frac{2}{\delta}}{2n}}.$$

