# OpenReview forum: "Generalization Analysis for Label-Specific Representation Learning"
_NeurIPS.cc/2024/Conference — NeurIPS 2024 spotlight_

### Official Review · Reviewer_BRgR · 2024-07-07

**Soundness:** 3
**Presentation:** 3
**Contribution:** 4
**Rating:** 8
**Confidence:** 4

**Summary:**

This paper proposes a new vector contraction inequality and derives a tighter label-specific representation learning (LSRL) generalization bound based on it.
This paper derives bounds for general function classes of LSRL with a tighter dependency on c than the SOTA, which provides a general theoretical guarantee for LSRL.And it also derives bounds for typical LSRL methods, which reveals the impact of different label- specific representations on the generalization analysis.

**Strengths:**

1. The paper proposes a new vector contraction inequality and derives a tighter LSRL generalization bound based on it, which has a more concise dependence on the number of labels compared to existing results.
2. The paper analyzes the generalization bounds of three typical LSRL methods, revealing the impact of different label-specific representations on generalization analysis, which helps better understand the good practical performance of LSRL.
3. The projection function class improves the vector-contraction inequalities  and decouples the relationship among different components.

**Weaknesses:**

The description of the derivation of the general bound in the main text is too brief. For better reading experience, the paper should give more details and some more intuitive description (such as lemma 1 and Theorem 1).

**Questions:**

The essence of LSRL is to divide multi-label classification into c binary classification problems. So why not refer to the binary classification problem's boundaries when deriving the boundary, such as k-means clustering-based LSRL method?

**Limitations:**

Yes.

---

> ### Author Rebuttal · Authors · 2024-08-07
>
> Thank you for your constructive comments and active interest in helping us improve the quality of the paper.
>
> The following are our responses to the Questions:
>
> **1. Response to the Weakness.**
>
> $\bullet$ We will add the proof sketch for Lemma 1 as follows:
>
> First, the Rademacher complexity of the loss function space associated with $\mathcal{F}$ can be bounded by the empirical $\ell_\infty$ norm covering number with the refined Dudley's entropy integral inequality. Second, according to the Lipschitz continuity w.r.t the $\ell_\infty$ norm, the empirical $\ell_\infty$ norm covering number of $\mathcal{F}$ can be bounded by the empirical $\ell_\infty$ norm covering number of $\mathcal{P}(\mathcal{F})$. Third, the empirical $\ell_\infty$ norm covering number of $\mathcal{P}(\mathcal{F})$ can be bounded by the fat-shattering dimension, and the fat-shattering dimension can be bounded by the worst-case Rademacher complexity of $\mathcal{P}(\mathcal{F})$. Hence, the problem is transferred to the estimation of the worst-case Rademacher complexity. Finally, we estimate the lower bound of the worst-case Rademacher complexity of $\mathcal{P}(\mathcal{F})$ by the Khintchine-Kahane inequality, and then combined with the above steps, the Rademacher complexity of the loss function space associated with $\mathcal{F}$ can be bounded.
>
> This proof sketch has been moved to the appendix due to the limitation of paper length.
>
> $\bullet$ We will add the proof sketch for Theorem 1 as follows:
>
> We first upper bound the worst-case Rademacher complexity $\tilde{\Re}_{nc}(\mathcal{P}(\mathcal{F}))$, and then combined with Lemma 1, the desired bound can be derived.
>
> We will add these proof sketches to the main paper in the revised version to improve the readability of the paper.
>
> **2. Response to the Question.**
>
> Since the prediction function of multi-label learning is a vector-valued function, which makes traditional Rademacher complexity analysis methods applicable to the class of scalar-valued functions or binary classification functions invalid. Hence, we need to convert the Rademacher complexity of a loss function space associated with the vector-valued function class $\mathcal{F}$ into the Rademacher complexity of a tractable scalar-valued function class or binary classification function class. We can then use existing analysis methods to bound the complexity of the class of scalar-valued functions or binary classification functions.
>
> A basic bound with a linear dependency on the number of labels (c) comes from the following inequality:
>
> $
> \mathbb{E}\left[\sup_{\boldsymbol{f} \in \mathcal{F}} \frac{1}{n} \sum_{i=1}^n \sum_{j=1}^c \epsilon_{ij} f_j\left(\boldsymbol{x}_{i}\right)\right]
> $
>
> $
> 	\leq  c \max_j \mathbb{E} \left[\sup_{f_j} \frac{1}{n} \sum_{i=1}^n  \epsilon_{ij} f_j\left(\boldsymbol{x}_{i}\right) \right]
> $,
>
> where we can use the generalization analysis method of the binary classification problem to bound the Rademacher complexity on the right side of the inequality. The dependency of the bounds on c can be improved to square-root by preserving the coupling among different components. However, these existing analysis methods based on preserving the coupling are not applicable to LSRL, and the induced generalization bounds are not tight enough and have a strong dependency on c. Hence, we introduce the projection function class to decouple the relationship for LSRL and derive new vector-contraction inequality, which convert the Rademacher complexity of a loss function space associated with the vector-valued function class $\mathcal{F}$ into the Rademacher complexity of the projection function class (i.e., binary classification function class), then we can derive tighter bounds with a weaker dependency on c than the SOTA.
>
> For k-means clustering-based LSRL method, when we bound the complexity of the projection function class, in the binary classification problem corresponding to each label, we use k-means clustering to generate the centers of the K clusters. Hence, the complexity of the function class of k-means clustering needs to be considered in bounding the complexity, as we discussed in Remark 2. When bounding the complexity of the function class of k-means clustering, we did not consider directly using existing analysis methods for clustering, since existing analysis methods often lead to bounds with a linear or stronger dependency on the number of clusters (K). Hence, in order to obtain a tighter generalization bound that matches the lower bound $\Omega(\sqrt{{K}/{n}})$ for k-means clustering, we formalize k-means clustering as a vector-valued learning problem, and develop a novel vector-contraction inequality for k-means clustering that can induce bounds with a square-root dependency on K. The generalization bound of k-means clustering-based LSRL method is tighter than the state of the art with a faster convergence rate $\tilde{O}(\sqrt{{K}/{n}})$, which is independent on c. Our bound is (nearly) optimal, up to logarithmic terms, both from the perspective of clustering and from the perspective of multi-label learning.

---

### Official Review · Reviewer_iWbs · 2024-07-08

**Soundness:** 4
**Presentation:** 3
**Contribution:** 4
**Rating:** 7
**Confidence:** 5

**Summary:**

The article discusses the theory bounds of Label-Specific Representation Learning (LSRL) in multi-label learning. It highlights the need for a deeper understanding of LSRL's generalization properties and proposes novel bounds based on Rademacher complexity. This paper derives bounds for general function classes of LSRL with a tighter dependency on the number of labels than the state of the art, and these theoretical results reveal the impact of different label-specific representations on generalization analysis.

**Strengths:**

1. This paper develops a novel vector-contraction inequality and derive the generalization bound for general function class of LSRL, which has not been explored in previous research.
2. By deriving tighter bounds and analyzing typical LSRL methods, the paper enhances theoretical guarantees and shed light on the impact of label-specific representations on generalization.
3. By exploring the effects of different label-specific representations, this paper provides insights into the generalization of LSRL and contributes to a deeper understanding of multi-label learning.
4. The assumptions posited in this paper are reasonable and aligned with real scenarios, with a logically coherent process of reasoning and argumentation.

**Weaknesses:**

1. Section 5.3 of this paper precisely assumes a fixed structure for the DNN-based LSRL method, which appears somewhat biased. It could be more comprehensive to indicate bounds for a subset of network structures sharing common characteristics.
2. Different label-specific representations exhibit substantial variations in their effect on Generalization Bounds. The paper solely undertakes theoretical analyses of a limited number of label-specific representations, potentially lacking comprehensiveness.
3. While the theoretical advancements are significant, the paper does not provide guidance on implementing LSRL in real-world applications.

**Questions:**

1. How can the theoretical bounds proposed in the study be translated into practical enhancements in real-world multi-label learning tasks?
2. To what extent can the assumptions in the theoretical analyses be relaxed to encompass a wider array of diverse and intricate problems? For instance, could these assumptions be extended to other loss functions?

---

> ### Author Rebuttal · Authors · 2024-08-07
>
> Thank you for your constructive comments and active interest in helping us improve the quality of the paper.
>
> **1. Response to the Weakness 1.**
>
> We appreciate the reviewer's suggestion for a broader scope of applicability of the analysis method for DNN-based LSRL. The analysis of the precise structure of the network is mainly affected by the specific model corresponding to the specific LSRL method being analyzed. However, the applicability of the analysis method can be relaxed, and the network structure corresponding to the theoretical result can be appropriately relaxed to a deep GCN connected to a deep feedforward neural network (FNN).
>
> The analysis of DNN-based LSRL here uses the capacity-based generalization analysis method. For deep models, the common capacity-based analysis method is to use a "peeling" argument, i.e., the complexity bound for l-layer networks is reduced to a complexity bound for (l-1)-layer networks. In this method, a product factor of the constant related to the Lipschitz property of the activation function and the upper bound of the norm of the weight matrix will be introduced for each reduction due to scaling. After applying the reduction l times, the multiplication of product factors with exponential dependency on l will make the bound vacuous. So far, effective capacity-based analysis for DNN (with a weak dependency on l) remains an open problem.
>
> However, the theoretical result in Section 5.3 is not negatively affected by l, since the network involved in GCN has only 3 layers, and the subsequent connected FNN has only 2 layers. These shallow networks do not make the bound vacuous, which is reflected by $D^5$ in bound.
>
> When considering a broader scope of applications, we have to consider deep GCNs and FNNs. If the "peeling" argument is used, it will bring concerns about l to bound. However, for deep GCNs, the increase in depth means the bound with a heavy dependency on l, which means that the performance will deteriorate, which is consistent with empirical performance, since in experiments, increasing the number of layers of GCN will lead to worse performance. Hence, there is still a need to develop new theoretical tools to analyze capacity-based bounds for deep FNNs and other deep network structures. We will focus on solving this open problem in future work.
>
> **2. Response to the Weakness 2.**
>
> We appreciate the reviewer's suggestion for the comprehensiveness. The main goal of this work is to address the deficiencies of existing theoretical results (i.e., existing bounds with a linear or square-root dependency on c cannot provide general theoretical guarantees for multi-label learning, and existing analysis methods that preserves the coupling are not applicable to LSRL), establish a theoretical framework and provide analysis tools for the generalization analysis of LSRL. Based on the proposed theoretical framework, to provide theoretical guarantees for existing LSRL methods as much as possible, we conduct analysis on several typical LSRL methods and provide new theoretical results and tools for k-means, Lasso, and DNNs. Although the LSRL methods analyzed here are limited, it is difficult for us to fully cover all LSRL methods with just one work. Our work fills the gap in theoretical analysis of LSRL, and in future work we will continue to expand the analysis of other LSRL methods to improve the comprehensiveness of theoretical analysis.
>
> **3. Response to Weakness 3 and Question 1.**
>
> First, existing theoretical results can improve the dependency of the basic bound on c from linear to square-root by preserving the coupling, which is reflected by the constraint $\\|\boldsymbol{w}\\| \leq \Lambda$. Preserving the coupling corresponds to high-order label correlations induced by norm regularizers. Our theoretical results for LSRL decouple the relationship among different components, and the bounds with a weaker dependency on c are tighter than the existing results that preserve the coupling, which also explains why LSRL methods outperform the multi-label methods that consider high-order label correlations induced by norm regularizers. Hence, when choosing the multi-label method in practice, one should prefer the LSRL methods rather than the latter.
>
> Second, the bound indicates that when we handle multi-label problems in practice, the method designed should ensure that the constant terms in the bound (other than c) are as small as possible. For example, when using DNN-based LSRL to handle multi-label problems, theoretical analysis shows that the bound is highly dependent on depth of GCN, which guides us not to design too many layers of GCN. In addition, the bound is linearly dependent on the maximum node degree of the label graph, which suggests that when the performance of the model is always unsatisfactory, we can check whether the maximum node degree is large. If so, we should consider using some techniques to remove some edges, e.g., DropEdge, to alleviate the over-fitting problem.
>
> **4. Response to the Question 2.**
>
> The assumption of Lipschitz continuity of the loss w.r.t. the $\ell_\infty$ norm is applicable to the theoretical analysis of other problem settings, such as multi-class classification, or more general vector-valued learning. Multi-class classification and multi-label learning are typical vector-valued learning problems. The assumption of Lipschitz continuity of the loss here is easy to satisfy. For multi-class classification, multi-class margin-based loss, multinomial logistic loss, Top-k hinge loss, etc. are all $\ell_\infty$ Lipschitz continuous. For multi-label learning, the surrogate loss for Macro-Averaged AUC is also $\ell_\infty$ Lipschitz continuous:
>
> $
> L_M (\boldsymbol{f}(\boldsymbol{x}, \boldsymbol{x}^\prime), \boldsymbol{y})
> $
>
> $
> =\frac{1}{c} \sum_{j=1}^c \ell \left(f_j(\boldsymbol{x})-f_j(\boldsymbol{x}^\prime)\right),
> $
>
> where $\boldsymbol{x}^{(\prime)}$ corresponds to the instances that are (ir)relevant to the $j$-th label.

---

### Official Review · Reviewer_xFN1 · 2024-07-11

**Soundness:** 3
**Presentation:** 3
**Contribution:** 3
**Rating:** 7
**Confidence:** 4

**Summary:**

This work makes one step towards the generalization analysis for label-specific representation learning. Compared with previous generalization studies on multi-label learning, this work provides a generalization bound with a much weaker dependency on the number of labels and decouple the relationship among components of different classes. This work also analyzes the generalization bounds for three typical LSRL methods, including k-means clustering-based method, LASSO-based method and DNN-based method, and these results reveal the impact of different label-specific representations.

**Strengths:**

1) Overall, I think this is a nice work with significant contribution. The proposed generalization bound takes a much weaker dependency on the number of labels, and this is quite important since a large number of labels may occur in the multi-label tasks. Besides, previous theoretical works for multi-label learning preserve the coupling among different components, and this is invalid for LSRL. This work decouples the relationship among different labels during the analysis, which is consistent with the process of LSRL. Therefore, I consider this work as a milestone for the theoretical understandings of LSRL methods.

2) The major technical contribution lies in the vector-contraction inequality given in Lemma 1, and the core idea is to introduce a projection function to decouple the relationship among labels. This may shed lights on relevant studies on LSRL. This work also analyzes the generalization bounds for three mainstream LSRL methods, and some techniques may be of independent interest for the studies of KNN, Lasso and deep neural networks.

3) This work is well written and the structure is nice. Both the motivations and contributions have been stated clearly. Necessary analysis has also been presented for the proposed theoretical results. Besides, a detailed review of related work is also provided to help readers understand the background of this work.

**Weaknesses:**

1）As pointed out in lines 165-168, the Rademacher complexity cannot be directly applied to the class of vector-valued functions F. Therefore, this work considers the complexity of the loss function space with respect to F. Here, I suggest formally defining the Rademacher complexity of this loss function space to avoid any misunderstanding.

2）This work provides definitions for the covering number and Fat-shattering dimension in Definition 2 and Definition 3. However, these complexities are not mentioned again in the main text. I suggest explaining how these complexities are used in the main text or directly moving their definitions to the appendix.

**Questions:**

1) In Eqn. (1), why does each label require defining two nonlinear mapping functions? Can one be used instead? Or could you please explain the different roles of these two non-linear functions?

2) As for the vector-contraction inequality given in Lemma 1, could it be applied to the analysis of other problems? For example, for multi-class classification, we sometimes convert the original problem into training multiple oneVsAll binary classifiers. Can the theoretical tools proposed in this article be used to explain the generalization of these methods?
3) Could the theoretical results explain why LSRL improves the performance of multi-label learning, compared with some traditional methods exploring the correlations among different labels?

**Limitations:**

See the above

---

> ### Author Rebuttal · Authors · 2024-08-07
>
> Thank you for your constructive comments and active interest in helping us improve the quality of the paper.
>
> **1. Response to Weakness 1.**
>
> As the reviewer said, formally defining the loss function space can avoid any misunderstanding. This problem is mainly caused by the limitation of paper length. We give the definition of the loss function space in Appendix and mix the two function classes with a description in the main paper. We will formally define the loss function space in the main paper:
> $$
> \mathcal{L}=\\{\ell (\boldsymbol{f}(\boldsymbol{x}), \boldsymbol{y}) : \boldsymbol{f} \in \mathcal{F}\\},
> $$
> where $\mathcal{F}$ is the vector-valued function class of LSRL defined in the main paper. Then, the empirical Rademacher complexity of the loss function space associated with $\mathcal{F}$ is defined by
>
> $\hat{\Re}_{D}(\mathcal{L})$
>
> $=\mathbb{E} \left[\sup_{\ell \in \mathcal{L}, \boldsymbol{f} \in \mathcal{F}}\left| \frac{1}{n} \sum_{i=1}^n \epsilon_{i} \ell \left(\boldsymbol{f} (\boldsymbol{x}_{i})\right)\right|\right]$.
>
> In addition, we will modify the relevant symbols accordingly, e.g., changing $\hat{\Re}_D(\mathcal{F})$ on the left side of the vector-contraction inequality in Lemma 1
>
> to $\hat{\Re}_{D}(\mathcal{L})$.
>
> **2. Response to Weakness 2.**
>
> These complexities are used to prove the vector-contraction inequality, the proof sketch is as follows:
>
> First, Rademacher complexity of the loss function space associated with $\mathcal{F}$ can be bounded by empirical $\ell_\infty$ norm covering number with refined Dudley's entropy integral inequality. Second, according to $\ell_\infty$ Lipschitz continuity, empirical $\ell_\infty$ norm covering number of $\mathcal{F}$ can be bounded by empirical $\ell_\infty$ norm covering number of $\mathcal{P}(\mathcal{F})$. Third, empirical $\ell_\infty$ norm covering number of $\mathcal{P}(\mathcal{F})$ can be bounded by fat-shattering dimension, and fat-shattering dimension can be bounded by worst-case Rademacher complexity of $\mathcal{P}(\mathcal{F})$. Hence, the problem is transferred to the estimation of worst-case Rademacher complexity. Finally, we estimate the lower bound of worst-case Rademacher complexity, and then the Rademacher complexity of the loss function space can be bounded.
>
> This proof sketch is moved to Appendix since the limitation of paper length. We will add it in the main paper to improve the readability.
>
> **3. Response to Question 1.**
>
> The purpose of 2 nonlinear mappings in definition is to emphasize the construction of label-specific representation (LSR). Since the goal of LSRL is to construct the representation with specific discriminative properties for each class label to facilitate its discrimination process, so the inner nonlinear mapping $\phi$ corresponds to the nonlinear transformation induced by the construction method of LSR, while the outer nonlinear mapping $\zeta$ refers to the nonlinear mapping corresponding to the classifier learned on the generated LSR. If only one nonlinear mapping is used to define the prediction function, it means learning a nonlinear classifier directly on the input data, which is the function class of general multi-label learning rather than LSRL.
>
> **4. Response to Question 2.**
>
> The vector-contraction inequality and the theoretical tools here are applicable to the analysis of other problem settings, such as multi-class classification, or more general vector-valued learning. Multi-class classification and multi-label learning are typical vector-valued learning problems. When applying the vector-contraction inequality and related theoretical results here, it is only necessary to check whether the loss function is $\ell_\infty$ Lipschitz continuous. In addition to the multi-label loss involved here, for multi-class classification, multi-class margin-based loss, multinomial logistic loss, Top-k hinge loss, etc. are all $\ell_\infty$ Lipschitz continuous.
>
> **5. Response to Question 3.**
>
> First, existing theoretical results can improve the dependency of the basic bound on c from linear to square-root by preserving the coupling, which is reflected by the constraint $\\|\boldsymbol{w}\\| \leq \Lambda$. Preserving the coupling corresponds to high-order label correlations induced by norm regularizers. Decoupling the relationship in LSRL is reflected by the constraint $\\|\boldsymbol{w}_j\\| \leq \Lambda$ for any $j \in [c]$.
>
> As a comparison, when $\\|\boldsymbol{w}_j\\|_2 \leq \Lambda$ for any $j \in [c]$,
>
> if we consider the group norm $\\|\cdot \\|_{2, 2}$,
>
> we have $\\|\boldsymbol{w}\\|_{2, 2} \leq \sqrt{c}\Lambda$, which means that these improved bounds still suffer from a linear dependency on c. Hence, the improvement in preservation of coupling by a factor of $\sqrt{c}$ benefits from replacing $\Lambda$ with $\sqrt{c}\Lambda$ in the constraint to some extent, and preserving the coupling corresponds to a stricter assumption. Our theoretical results decouple the relationship, and the bounds with a weaker dependency on c are tighter than existing results that preserve the coupling, which also explains why LSRL methods outperform multi-label methods that consider high-order label correlations induced by norm regularizers.
>
> Second, based on our theoretical results, we can find that LSRL methods substantially increase the data processing, i.e., the process of constructing LSRs. From the perspective of model capacity, compared with traditional multi-label methods, since the introduction of construction methods of LSR, the capacity of the model is significantly increased, especially if deep learning methods are used to generate LSRs, which improves the representation ability of the model to a certain extent. Or more intuitively, LSRL means an increase in model capacity and stronger representation ability, which makes it easier to find the hypotheses with better generalization in function class. Hence, it is reasonable that LSRL can improve the performance of multi-label learning.

---

> > ### Comment · Reviewer_xFN1 · 2024-08-13
> >
> > Thank you for the author's detailed response. My issue has been resolved, and I will keep my score.

---

> > > ### Author Response · Authors · 2024-08-13
> > >
> > > Thank you again for your support.

---

### Official Review · Reviewer_Axy4 · 2024-07-12

**Soundness:** 2
**Presentation:** 3
**Contribution:** 2
**Rating:** 5
**Confidence:** 4

**Summary:**

The paper focuses on the theoretical analysis of Label-Specific Representation Learning (LSRL) within the context of multi-label learning. LSRL aims to improve multi-label learning by creating representations with distinct discriminative properties for each label. While LSRL has shown empirical success, its theoretical underpinnings, especially regarding generalization bounds, have been less explored. The authors propose a novel vector-contraction inequality and derive tighter generalization bounds for LSRL. These bounds offer a better dependency on the number of labels compared to existing theories. The paper also discusses the implications of these bounds for various LSRL methods, emphasizing the mild assumptions that explain the good generalization abilities of LSRL.

**Strengths:**

1. Novel Theoretical Contributions: The paper introduces a new vector-contraction inequality specific to LSRL, providing a theoretical framework that was previously lacking.
2. Improved Generalization Bounds: The derived generalization bounds have a weaker dependency on the number of labels, offering better theoretical guarantees.

**Weaknesses:**

1. The main issue addressed in the paper is viewing the multi-label problem as multiple binary classifications and then identifying the most discriminative features for each label. However, it is well known that the key aspect of multi-label problems lies in label correlations, which are crucial for improving the effectiveness of the methods. If a model considers each label independently and ignores label correlations, it is no different from solving a multi-class classification problem. This paper does not consider label correlations, raising questions about whether the proposed theoretical framework can substantially enhance multi-label learning methods.
2. It is well known that in the era of big data, multi-label problems often face the issue of extreme multi-labels, where the scale of labels is very large. In such cases, implementing the LSRL framework is nearly impossible due to its high complexity. Therefore, although the authors propose better generalization bounds, the computational cost required to achieve this generalization is extremely high. The paper lacks an analysis of the complexity of multi-label methods under the LSRL framework. The authors are requested to provide further analysis in this regard.

**Questions:**

1. The paper seems somewhat unclear when emphasizing the innovative aspects of its proof process. Could the authors clearly articulate the unique innovations in their proof of the generalization bounds and explain how their approach differs from and improves upon the generalization proofs of other multi-label methods?
2. Could the authors provide the key theoretical tools and intermediate steps involved in the  derivation steps of the vector-contraction inequality?
3. For the k-means clustering-based LSRL method, how does the two-stage approach affect the derivation of generalization bounds? Specifically, how do the clustering center generation and label-specific representation generation stages interact in the derivation process?

**Limitations:**

Theory should serve practice or guide the design of better algorithms. However, this paper lacks experimental validation of the theory, particularly including a comparison of the effectiveness of multi-label methods under the LSRL framework with other methods that have larger generalization errors. The paper also lacks numerical validation of the theory and verification of whether the theoretical assumptions are easily met.

---

> ### Author Rebuttal · Authors · 2024-08-07
>
> Thank you for your constructive comments and active interest in helping us improve the quality of the paper.
>
> **1. Response to Weakness 1.**
>
> a. The basic idea of LSRL is to decompose multi-label problem into c binary classification problems. This idea is effective for handling multi-label problems, since Binary Relevance is a popular and important multi-label method.
>
> b. Label correlations mainly focus on the processing of label space by exploiting relationships between labels, while decomposition-based LSRL focuses on the operation on input space and implicitly considers label correlations in the process of constructing label-specific representations (LSR). For example, in construction of LSR in LLSF and CLIF, label correlation information is embedded into LSR in input space by introducing the sharing constraint or using a GCN over the label graph to generate label embeddings. LSRL is more effective since label correlation information is considered in construction of LSR.
>
> **2. Response to Weakness 2.**
>
> It is not a trivial problem to extend existing multi-label methods to handle extreme multi-label (EML) problems. For the treatment of EML problems, it is often necessary to introduce specific strategies in label space. Existing LSRL methods have not considered handling EML situations. LSRL needs to combine some specific strategies to deal with EML problems, which is more of an algorithm design problem. The main goal of our work is to provide an effective theoretical framework and analysis tools. After exploring an effective specific strategy for LSRL, we can formally define the strategy (i.e., characterize it more precisely in function class) and conduct further detailed analysis in combination with the theoretical framework provided here.
>
> **3. Response to Question 1.**
>
> The analysis of multi-label learning can be traced back to a basic bound with a linear dependency on the number of labels (c), which comes from the following inequality:
>
> $
> \mathbb{E}\left[\sup_{\boldsymbol{f}\in\mathcal{F}}\frac{1}{n}\sum_{i=1}^n\sum_{j=1}^c\epsilon_{ij}f_j\left(\boldsymbol{x}_{i}\right)\right]
> $
>
> $
> \leq c\max_j\mathbb{E}\left[\sup_{f_j}\frac{1}{n}\sum_{i=1}^n\epsilon_{ij}f_j\left(\boldsymbol{x}_{i}\right)\right]
> $.
>
> The dependency of bounds on c can be improved to square-root. Such improvements essentially come from preserving the coupling reflected by the constraint $\\|\boldsymbol{w}\\|\leq\Lambda$.
>
> As a comparison, when $\\|\boldsymbol{w}_j\\|_2\leq\Lambda$ for any $j\in[c]$,
>
> if we consider group norm $\\|\cdot\\|_{2, 2}$,
>
> we have $\\|\boldsymbol{w}\\|_{2, 2}\leq\sqrt{c}\Lambda$, which means that these improved bounds still suffer from a linear dependency on c.
>
> Hence, the improvement in preservation of coupling by a factor of $\sqrt{c}$ benefits from replacing $\Lambda$ with $\sqrt{c}\Lambda$ in the constraint to some extent. The bounds can be improved to be independent on c for $\ell_\infty$ Lipschitz loss by preserving the coupling. However, these theoretical results based on preserving the coupling do not apply to LSRL. Hence, we introduce the projection function class to decouple the relationship for LSRL and derive new vector-contraction inequality, which lead to tighter bounds than the SOTA. Other theoretical innovations on typical LSRL methods mainly include: new vector-contraction inequality for k-means that can induce bounds with a weaker dependency on K, introduction of pseudo-Lipschitz function, and generalization analysis of GCN, as we discussed in Remarks.
>
> **4. Response to Question 2.**
>
> First, Rademacher complexity of the loss function space associated with $\mathcal{F}$ can be bounded by empirical $\ell_\infty$ norm covering number with refined Dudley's entropy integral inequality. Second, based on $\ell_\infty$ Lipschitz continuity, empirical $\ell_\infty$ norm covering number of $\mathcal{F}$ can be bounded by empirical $\ell_\infty$ norm covering number of $\mathcal{P}(\mathcal{F})$. Third, empirical $\ell_\infty$ norm covering number of $\mathcal{P}(\mathcal{F})$ can be bounded by fat-shattering dimension that can be bounded by worst-case Rademacher complexity of $\mathcal{P}(\mathcal{F})$. Hence, the problem is transferred to the estimation of worst-case Rademacher complexity. Finally, we estimate the lower bound of worst-case Rademacher complexity by Khintchine-Kahane inequality, and then Rademacher complexity of the loss function space can be bounded.
>
> **5. Response to Question 3.**
>
> In k-means clustering-based LSRL, for each label, we first use k-means to get centers of K clusters, and then we use K centers to construct LSR. We cannot formally express the two steps in a closed-form expression by a composite function since K centers are generated by arg min function. A common and trivial analysis method for two-stage methods is to start analysis directly from second stage, thereby simplifying the difficulty of theoretical analysis caused by the difficulty of formalization in first stage, which actually ignores the complexity of the method in first stage.
>
> Here, K centers generated in first stage are actually used as fixed parameters rather than inputs in second stage. Hence, to fully consider the model capacity corresponding to first stage, it is reasonable to define the function class as sum of classes $\mathcal{F} + \mathcal{G}$ corresponding to the methods of two stages. Then, combined with novel vector-contraction inequality, the analysis is transformed into bounding the complexity of projection function class $\mathcal{P}(\mathcal{F}+\mathcal{G})$. Benefiting from the structural result of complexity, we can bound the complexity of $\mathcal{P}(\mathcal{F})$ and $\mathcal{P}(\mathcal{G})$ separately, where bounding $\mathcal{P}(\mathcal{F})$ is obvious. To bound $\mathcal{P}(\mathcal{G})$, we develop a novel vector-contraction inequality for k-means that can induce bounds with a square-root dependency on K, as we discussed in Remark 2 and line 623-627.

---

> ### Author Response · Authors · 2024-08-12
> **Kind Reminder to Reviewer Axy4**
>
> Dear reviewer,
>
> Thank you for your insightful feedback and constructive comments! We have answered and explained your questions in the rebuttal (along with the global rebuttal). We are eager to hear back from you if you have any feedback or further questions, and we would love to know your updated reviews.
>
> Regards,
>
> Authors

---

### Author Rebuttal · Authors · 2024-08-07

We would like to thank all reviewers for your efforts in reviewing this paper, as well as your constructive comments and active interest in helping us improve the quality of the paper.

We provide detailed responses to the questions of each reviewer. Here we first make a brief summary of our responses:

**$\bullet$ On the derivation process of the novel vector-contraction inequality (i.e., Lemma 1):** The proof sketch for Lemma 1 is moved to the appendix due to the limitation of paper length. We will add the corresponding proof sketches to the main paper to improve the readability of the paper.

**$\bullet$ Explanations and discussions of problems such as the wider applicability of theoretical results and their guidance for practice:** We have provided detailed responses to each reviewer's specific questions, and we will add a Discussion section to incorporate these explanations and discussions of these problems into the paper in an appropriate manner.

----

Next, since the limitation of 6000 characters in each rebuttal, we will provide detailed response to the limitation raised by Reviewer Axy4 in this global rebuttal.

**$\bullet$ Response to the Limitation by Reviewer Axy4**

**(1)** This is a purely theoretical work on capacity-based generalization analysis. There was no theoretical analysis on LSRL before. The main goal of this work is to address the deficiencies of existing theoretical results (i.e., existing bounds with a linear or square-root dependency on c cannot provide general theoretical guarantees for multi-label learning, and existing generalization analysis methods that preserves the coupling among different components are not applicable to LSRL), establish a theoretical framework and provide effective and general theoretical analysis tools for the generalization analysis of LSRL. Based on the proposed theoretical framework, we focus on the capacity of models corresponding to several typical LSRL methods, analyze the complexity of these LSRL models, obtain capacity-based generalization bounds, and provide new theoretical results and tools for k-means clustering, Lasso, and DNNs. Compared with algorithm-based generalization analysis, experimental verification is not necessary for capacity-based generalization analysis. The style of this type of purely theoretical work is similar, as in literature [1-9], etc.

[1] Tomer Levy, Felix Abramovich. "Generalization Error Bounds for Multiclass Sparse Linear Classifiers", JMLR 2023.

[2] Yi-Fan Zhang, Min-Ling Zhang. "Nearly-tight Bounds for Deep Kernel Learning", ICML 2023.

[3] Yunwen Lei, Tianbao Yang, Yiming Ying, Ding-Xuan Zhou. "Generalization Analysis for Contrastive Representation Learning", ICML 2023.

[4] Shaojie Li, Yong Liu. "Sharper Generalization Bounds for Clustering", ICML 2021.

[5] Peter L. Bartlett, Nick Harvey, Christopher Liaw, Abbas Mehrabian. "Nearly-tight VC-dimension and Pseudodimension Bounds for Piecewise Linear Neural Networks", JMLR 2019.

[6] Noah Golowich, Alexander Rakhlin, Ohad Shamir. "Size-Independent Sample Complexity of Neural Networks", COLT 2018.

[7] Andreas Maurer. "Bounds for Linear Multi-Task Learning", JMLR 2006.

[8] Sara van de Geer. "On Tight Bounds for the Lasso", JMLR 2018.

[9] Nathan Srebro, Karthik Sridharan, Ambuj Tewari. "Smoothness, Low Noise and Fast Rates", NIPS 2010.

**(2) On the rationality and satisfiability of assumptions.** In fact, the mild assumptions here are easy to satisfy. The only assumptions involved in our analysis are the boundedness of the functions and the Lipschitz continuity of the loss functions. These assumptions are relatively mild and are satisfied in practical applications. As we explain in the main paper, the $\ell_\infty$ Lipschitz continuity of the loss functions is satisfied for the commonly used loss functions we analyzed, as proved by Proposition 1.

The assumption of the boundedness of the functions is a common assumption in generalization analysis for various learning settings, such as deep learning [2], [6], clustering [4], kernel methods [10], [11], [12], multi-classification [13], etc. In addition, in practice, the boundedness of functions can often be guaranteed, which appears in empirical evidences as support for constraints (i.e., boundedness) on weights and nonlinear mappings.

For shallow models, such constraints are easy to satisfy. Since normalization of input features is a common data preprocessing operation in machine learning, performing single nonlinear mapping on the normalized data will not significantly change the scale of the data, which ensures the boundedness of nonlinear mappings. Through proper data preprocessing and normalization, the scale of weights of the final model with good performance is controllable, i.e., the numerical values of weights are bounded.

For deep models, in order to achieve good generalization performance, many strategies and techniques commonly used in practice are actually to ensure that these constraints hold. For example, in order to prevent exploding gradient in the training of DNNs, commonly used techniques such as weight regularization and Xavier initialization are to ensure that the weights are within a certain range (i.e., the boundedness of weights), and commonly used techniques such as mean normalization and batch normalization are to ensure that the inputs and outputs of each layer are within a certain range (i.e., the boundedness of nonlinear mappings).

[10] Corinna Cortes, Mehryar Mohri, Afshin Rostamizadeh. "Generalization Bounds for Learning Kernels", ICML 2010.

[11] Corinna Cortes, Marius Kloft, Mehryar Mohri. "Learning Kernels Using Local Rademacher Complexity", NIPS 2013.

[12] Nathan Srebro, Shai Ben-David. "Learning Bounds for Support Vector Machines with Learned Kernels", COLT 2006.

[13] Yunwen Lei, Ürün Dogan, Ding-Xuan Zhou, Marius Kloft. "Data-dependent generalization bounds for multi-class classification", IEEE Transactions on Information Theory 2019.

---

### Decision · Program_Chairs · 2024-09-25

**Decision:**

Accept (spotlight)

**Comment:**

The reviewers of this paper provided thorough reviews and engaged with the authors during the discussion period. The authors propose a new (tighter) bound for label-specific representation learning based on a novel vector-contraction inequality. The reviewers have praised the work and have been satisfied with the answers given by the authors during the feedback period.